# Establishing functional giant *Dictyostelium* cells reveals front–rear polarity in intracellular signaling
Yukihisa Hayashida[1], Yuki Gomibuchi[2], Chikoo Oosawa [2], Takuo Yasunaga[2] & Yusuke V. Morimoto [2] ✉

Intracellular signaling dynamics are often obscured by the spatial and temporal limitations of cell size. Here, we developed a method to enlarge **Dictyostelium discoideum** cells by partial cytokinesis inhibition, generating multinucleated yet functional giant cells. These cells retained chemotactic signaling, polarity, and motility, enabling high-resolution live-cell imaging. Using fluorescent probes for cAMP and $Ca^{2+}$, we uncovered a directional, front-to-rear propagation of cAMP signaling and a biphasic $Ca^{2+}$ response coordinated with actin wave dynamics. Grid-based mapping revealed asymmetric cAMP synthesis and decay kinetics, and vesicle localization suggested spatially regulated cAMP secretion. Our findings demonstrate that intracellular signaling involves self-organized, spatially structured propagation events aligned with cellular polarity. The giant cell platform offers a powerful and generalizable strategy for dissecting the spatiotemporal logic of single-cell signaling.

In multicellular systems, individual cells exchange information through intercellular signaling, enabling diverse biological phenomena such as immune responses and morphogenesis[1–4]. These intercellular signaling events are ultimately governed by intracellular signal transduction. Therefore, elucidating the mechanisms of intracellular signaling is fundamental to understanding signal integration at the organismal level.

The social amoeba *Dictyostelium discoideum*, the model organism in this study, lives in a unicellular state by feeding on bacteria, but when starved, tens of thousands of cells assemble to form a multicellular body by using cyclic adenosine monophosphate (cAMP) as a chemoattractant[5–8]. In this process, signal reception and production of cAMP in response to starvation coordinate large-scale cell aggregation, making *Dictyostelium* a well-established model for signal transduction studies[9–11]. *Dictyostelium* cells allow for robust experimental manipulation, including genetic modification and live-cell imaging with fluorescent biosensors. Upon cAMP stimulation, cells synthesize and secrete cAMP in a relay fashion, accompanied by periodic fluctuations in cytosolic $Ca^{2+}$ levels[6,7,12–14].

Despite extensive studies, the spatiotemporal dynamics of intracellular signaling events remain poorly understood due to the small sizes of cells. Fluorescent biosensors that change emission wavelength in response to signaling molecules can be introduced to monitor these dynamics[15–18], however, high-speed imaging requires short exposure times, which, in combination with phototoxicity constraints, leads to low signal-to-noise

ratios and reduced measurement accuracy. Given the well-developed genetic tools and databases available for *Dictyostelium*[19,20], single-cell signal measurement in this organism is a crucial yet technically demanding task due to its relatively small cell size. Compounding this issue, the diffusion coefficient of cAMP in the cytosol ranges from 136 to 780 $\mu m^2/s$[21], meaning that in *Dictyostelium* cells with surface areas of approximately $100\,\mu m^2$[22], cAMP can equilibrate across the cell in under a second. Under typical acquisition conditions, such rapid diffusion appears merely as blinking events[7], preventing accurate localization of signal propagation[23].

Although super-resolution microscopy has significantly improved spatial resolution, most methods are optimized for the imaging of structural components and are ill-suited for capturing the rapid diffusion of unstructured signaling molecules[24–26]. Expansion microscopy, which enhances spatial resolution by physically enlarging fixed cells and tissues, cannot be applied to live-cell signaling analysis[27–30].

Cell enlargement through electrofusion has been demonstrated in *Dictyostelium*[31,32], however, such enlarged cells are transient and incapable of intercellular signaling during development. While previous studies on cytokinesis inhibition have described multinucleated giant *Dictyostelium* cells[33–37], the signaling competence of such cells has not been systematically examined.

In this study, we developed a method to enlarge *Dictyostelium* cells by partially inhibiting cytokinesis while retaining intercellular signaling

[1]Graduate School of Computer Science and Systems Engineering, Kyushu Institute of Technology, Iizuka, Fukuoka, Japan. [2]Department of Physics and Information Technology, Faculty of Computer Science and Systems Engineering, Kyushu Institute of Technology, Iizuka, Fukuoka, Japan. ✉e-mail: yvm001@phys.kyutech.ac.jp

competence and cellular polarity. This approach dramatically improves spatiotemporal resolution for live-cell imaging of signal transduction.

## Results

### Generating giant *Dictyostelium* cells while preserving intercellular signaling competence

To investigate the mechanistic details of intracellular signaling events, we established a method to generate giant *D. discoideum* cells while retaining critical cellular functions, including intercellular signal relay. In *Dictyostelium*, cytokinesis requires both the contractile action of the actomyosin ring, mainly driven by actin and myosin II, and substrate adhesion-dependent traction forces[37,38]. When both mechanisms are compromised, cytokinesis fails, leading to the formation of multinucleated giant cells, as previously reported[33,36,37].

In particular, *mhcA paxB* double-knockout (KO) cells, which lack the myosin II heavy chain (*mhcA*) and the focal adhesion component Paxillin (*paxB*) genes, efficiently generate giant, multinucleated cells[37]. However, permanent gene disruptions may cause pleiotropic effects and compromise other signaling functions. To minimize such unintended effects, we systematically compared genetic knockouts with pharmacological and physical perturbations.

To inhibit contractile ring function, we used blebbistatin, a myosin II ATPase inhibitor. To disrupt adhesion-mediated traction forces, we cultured cells in suspension with gentle shaking. Consistently with previous studies, inhibition of either contractile activity or substrate adhesion alone did not significantly alter the cell-size distribution compared to that of wild-type AX2 cells under static conditions (Fig. 1a, b, e, g)[35,37]. When wild-type cells were cultured in suspension by shaking, an increase in slightly larger cells was observed, indicating that suspension culture is more effective than

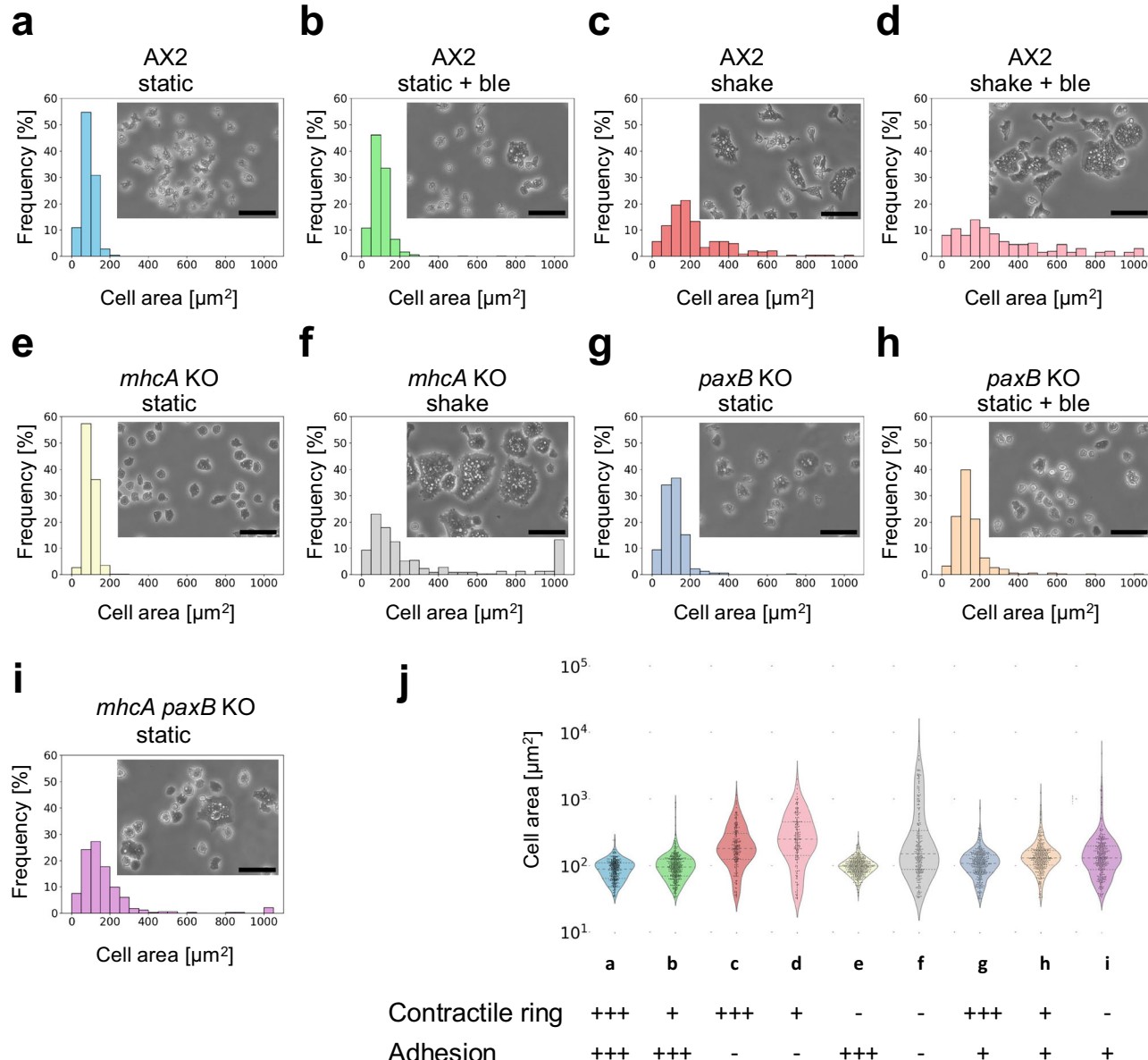

**Fig. 1 | Comparison of cell size distributions under different conditions of cytokinesis inhibition. a–i** Phase-contrast images and corresponding histograms showing cell size distributions in *Dictyostelium* cells cultured under various conditions: static culture (static) with or without blebbistatin (ble), and shaking culture (shake) in wild-type (AX2) (**a–d**), myosin II heavy-chain-knockout (*mhcA* KO) (**e**, **f**), *paxB*-knockout (*paxB* KO) (**g**, **h**), and double-knockout (*mhcA paxB* KO) (**i**) backgrounds. Cells were plated at $5 \times 10^5$ cells/mL on glass-bottom dishes, allowed to adhere, washed with DB buffer, and imaged using phase-contrast microscopy (scale bars: 50 μm). Histograms represent pooled data from three fields of view. **j** Summary of cytokinesis functionality under each condition: contractile ring (+++ intact, + partial inhibition, − complete loss); adhesion (+++ intact, + partial loss [*paxB* KO], − loss [shaking culture]).

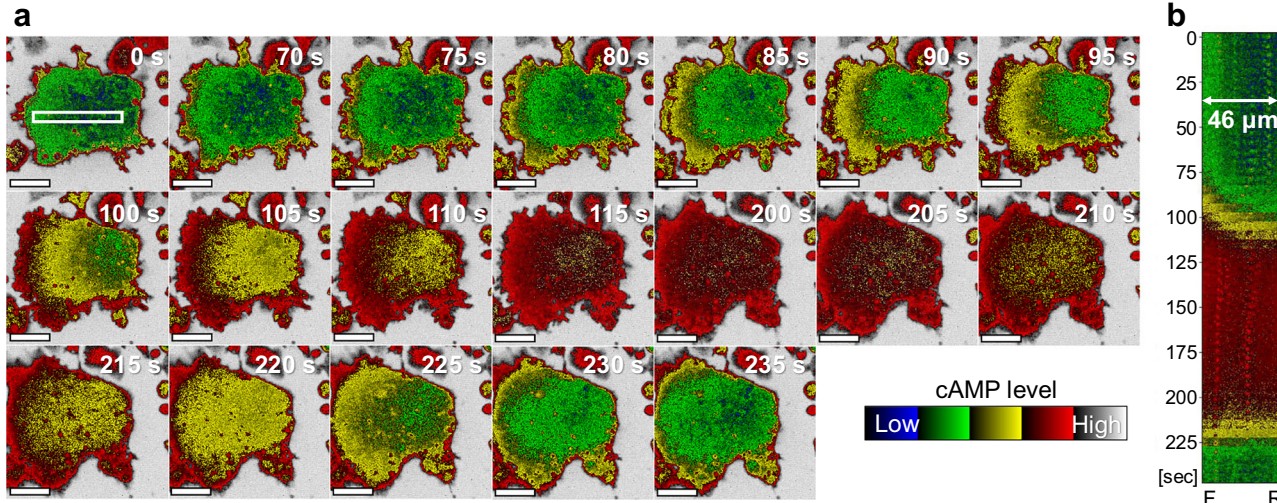

**Fig. 2 | cAMP signal propagation in a chemotactic giant cell. a** Time-lapse fluorescence images of Flamindo2-expressing giant cells after starvation for 7 h in DB showing spatial changes in cytoplasmic cAMP levels (inverted intensity). Fluorescence imaging was performed using a confocal microscope with a 40× objective. Cells were observed at 5 s intervals. The elapsed time from the start of imaging was shown. Scale bars: 20 μm. **b** Kymograph of cAMP signal intensity extracted from the white rectangle in (**a**) (46 μm in length), showing an earlier increase at the front (F) and an earlier decrease at the rear (R) over time.

myosin inhibition in promoting cell enlargement (Fig. 1c). In contrast, simultaneous disruption of both processes resulted in a marked increase in cell size, producing multinucleated giant cells not observed under normal conditions (Fig. 1d, f, h–j).

Simulations of cell-size distribution showed that varying the probability of cytokinesis yielded results consistent with experimental data, suggesting that giant cells arise stochastically through inhibition of cytokinesis rather than representing a distinct subpopulation caused by mutation (Supplementary Fig. 1).

Notably, the combination of blebbistatin treatment with suspension culture yielded enlarged multinucleated cells without genetic modifications (Fig. 1d). Whereas normally cultured cells exhibited a median area of 86.5 μm², only 0.5% of cells under these conditions exceeded 200 μm², a size more than twice the median (Fig. 1a). In contrast, under the conditions used in Fig. 1d, 59.5% of cells were larger than 200 μm², and 20.5% exceeded 500 μm², indicating that giant-cell formation was achieved with markedly higher efficiency. This condition was therefore selected for subsequent experiments, as it effectively generates giant cells while minimizing perturbations to endogenous signaling pathways.

### Giant cells retain intercellular cAMP relay and chemotactic responsiveness

To verify whether the giant cells retained functional intercellular signaling capabilities, we monitored their responses to cAMP using the genetically encoded fluorescent cAMP probe Flamindo[2,7,39]. In *Dictyostelium*, cells relay cAMP signals in response to starvation by cyclically sensing, synthesizing, and secreting cAMP[40–42]. This signaling relay drives chemotactic aggregation of approximately 10⁵ cells and occurs with a periodicity of approximately 6–7 min[7,43–45].

Flamindo2-expressing giant cells exhibited clear cyclic fluctuations in fluorescence intensity with a period of approximately 7 min, matching those of normal-sized cells within the same field of view (Supplementary Fig. 2a). This indicates that giant cells are capable of responding to extracellular cAMP pulses in a temporally coordinated manner. Furthermore, the giant cells also exhibited positive chemotactic migration toward cAMP, which appeared broadly similar to that of control cells (Supplementary Video 1). Additionally, we observed Ca²⁺ oscillations temporally coupled to cAMP relay events in the giant cells using the fluorescent Ca²⁺ probe GCaMP6s[46], which was consistent with prior reports on intact normal-sized cell populations (Supplementary Fig. 2b and Video 2)[14]. Together, these results demonstrate that the giant *Dictyostelium* cells retained essential signaling

functions, including intercellular cAMP relay, chemotaxis, and Ca²⁺ dynamics, comparable to those of normal-sized cells.

To further evaluate whether these giant cells can establish front–rear polarity during chemotaxis, we investigated the localization of the pleckstrin homology domain from Akt/PKB (PH$_{Akt}$), which accumulates at the leading edge of cells in response to cAMP stimulation. In *D. discoideum*, the PH$_{Akt}$ domain specifically binds to phosphatidylinositol (3,4,5)-trisphosphate (PIP$_3$), which accumulates at the front of the cell during directional migration[47].

To visualize PIP$_3$ dynamics, we expressed a PH$_{Akt}$-RFP fusion protein in the giant cells and monitored its subcellular distribution during chemotactic migration. As shown in Supplementary Fig. 2c–e, PH$_{Akt}$-RFP localized predominantly at the leading edge of the giant cells, mirroring the polarization observed in normal-sized cells. These results indicate that giant cells not only retain signal transduction capacity but also maintain front–rear cell polarity during chemotactic migration.

### Subcellular cAMP propagation can be observed in giant cells

Having confirmed that the giant cells retained signaling, motility, and polarity functions comparable to those of normal-sized cells (Supplementary Fig. 2), we next investigated whether the enlarged cell volume permitted visualization of subcellular variations in cAMP levels otherwise difficult to resolve in normal-sized cells. Previous studies using highly sensitive cAMP biosensors in *Dictyostelium* have enabled single-cell analyses but failed to resolve spatial cAMP gradients within individual cells[7,23].

Time-lapse imaging of a chemotactic giant cell expressing Flamindo2 revealed clear front-to-rear cAMP propagation not visible in normal-sized cells (Fig. 2). Between 70 and 115 s after the start of imaging, cAMP levels increased as the signal propagated from the front to the rear end of the migrating giant cell. Conversely, cAMP levels decreased between 205 and 235 s after the start of imaging as the signal propagated from the rear to front (Fig. 2 and Supplementary Video 3).

To quantitatively assess the correlation between intracellular cAMP dynamics and cell movement, we first approximated the migration direction by fitting the cell centroid trajectory (Fig. 3a) to a linear function, obtaining a slope of 0.42 (Fig. 3b and Supplementary Fig. 3a). Based on this vector, we defined the front and rear positions as 20 μm offsets along the fitted migration axis (x = −20, y = −8.4 for front; x = +20, y = +8.4 for rear). Fluorescence intensity changes at these locations were plotted over time (Fig. 3c). The resulting time series showed that cAMP level increases initiated at the front and decreases began earlier at the rear.

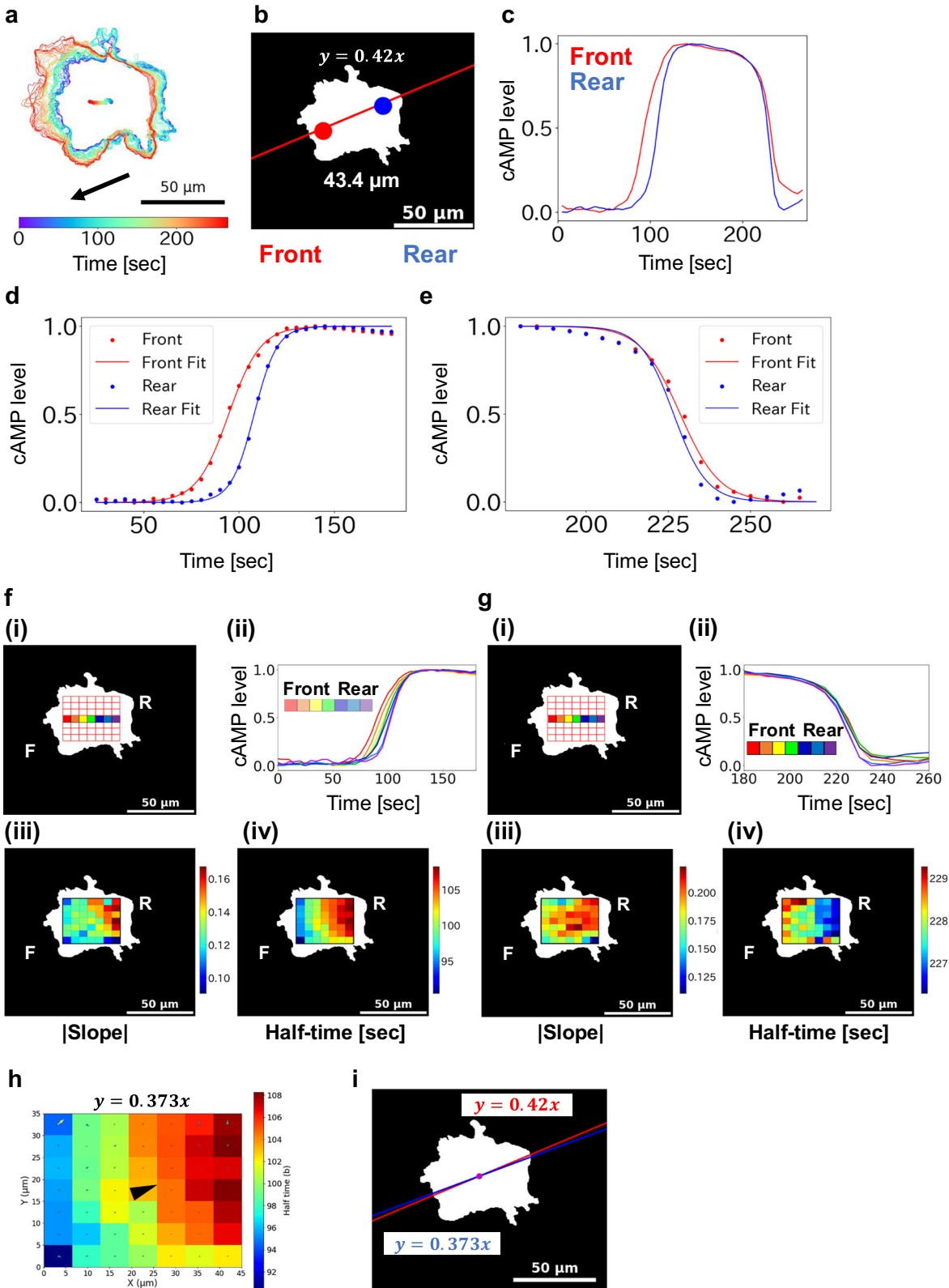

To extract kinetic parameters, the fluorescence changes were fitted to sigmoid functions (Fig. 3d, e). The slope of the fits indicated that cAMP accumulation and clearance occurred more rapidly at the rear than at the front (Supplementary Table S1). Assuming a wave-like propagation model with the slope of cAMP gradient, we estimated the average one-dimensional propagation velocity to be $2.98 \pm 0.31$ μm/s and the apparent diffusion

coefficient to be $63.9 \pm 10.0$ μm$^2$/s (Supplementary Table S2), based on three independent measurements (Fig. 3 and Supplementary Figs. 3–5).

The analyses in Fig. 3d, e focused on cAMP dynamics at the front and rear regions of the giant cells. However, these data do not address the possibility of spatial heterogeneity within each region, for example, whether there are localized differences even within the front domain, or how cAMP

**Fig. 3 | Analysis of cAMP signal propagation in a giant cell.** Representative analysis of the giant cell presented in Fig. 2. **a** Trajectory of the cell centroid during migration. The contour and centroid of the giant cell are overlaid. Arrow indicates the direction of migration. **b** Definition of front and rear regions for analysis. The trajectory of the centroid was fitted to a linear regression (slope = 0.42). Front and rear points were defined as 20 μm in the x-direction and ±0.42 × 20 μm in the y-direction from the centroid. Red and blue circles indicate the front and rear measurement regions, respectively (diameter: 11 μm). **c** Temporal changes in cAMP levels at the front (red) and rear (blue) regions. The cAMP level was calculated by inverted normalization. **d** Sigmoid fitting of cAMP increase at the front (red) and rear (blue) regions. **e** Sigmoid fitting of cAMP decrease using the same equation and color code as in (**d**). **f** Region for high-resolution analysis of intracellular cAMP increase. **i** A 45 × 35 μm rectangular area was divided into a 7 × 7 grid (each grid approximately 6.4 × 5 μm), as shown by the red box. **ii** Time-course of cAMP level during increase across the central horizontal row (row 4) of grids, color-coded corresponding to (**i**). **iii** Heatmap of the absolute slope parameter from sigmoid fitting of cAMP increase in each grid. **iv** Heatmap of the half-time parameter from the same fitting in (**ii**). **g** Region for high-resolution analysis of cAMP decrease, as in (**f**), for the same cell. **h** Heatmap of the half-time parameter of cAMP increase for determining the intracellular cAMP gradient direction. Δx = 6.429 μm; Δy = 5.000 μm. Each arrow represents a local gradient vector, and the central arrow indicates the representative overall gradient slope. **i** Comparison of cell migration direction with cAMP increase gradient direction. Red: cell migration direction; Blue: cAMP increase gradient.

levels behave at the cell center. To address this, we divided each giant cell into a 7 × 7 grid, where each grid unit approximately corresponded in area to a normal-sized *Dictyostelium* cell, and analyzed cAMP signal kinetics in each grid unit (Fig. 3f, g).

Analysis of cAMP increase dynamics along the central row provided stronger support for a wave-like propagation (Fig. 3f(ii)), with the signal initiating in the lower-left region of the cell (relative to the image frame) and spreading throughout the entire cell with progressively steeper slopes.

In contrast, cAMP signal decay showed a less pronounced spatial gradient (Fig. 3g(ii)), but consistently initiated in the rear of the cell before the front, again confirming directionally biased cAMP clearance. Across all recordings, the decay slope was steeper in the rear than in the front, indicating a faster rate of signal clearance in the rear compartment (Fig. 3g, Supplementary Figs. 3–5 and Table S3).

Importantly, the heatmaps derived from each grid revealed that the direction of the cAMP synthesis gradient aligned with the direction of cell migration (Fig. 3h, i and Supplementary Video 3). Given that *Dictyostelium* cells exhibit chemotaxis toward extracellular cAMP, this observation suggests that cells may locally initiate cAMP synthesis at the site of signal reception, thereby promoting directional migration via intracellular reinforcement of chemotactic cues.

To eliminate potential artifacts arising from probe expression heterogeneity and cell movement, we used a ratiometric sensor, Flamindo2-mRFPmars, which combines the cAMP-responsive fluorescence of Flamindo2 with a cAMP-insensitive red fluorescent protein. By calculating RFP/Flamindo2 ratios, we corrected intensity changes for variations in probe concentration and changes in cell shape due to movement. In these cells, cAMP levels increased from the leading edge and decreased from the rear (Supplementary Figs. 6–9). These dynamics, independent of cell shape or expression artifacts, confirm intrinsic intracellular cAMP propagation. Sigmoid fitting revealed a steeper slope for decay in the rear than in the front, indicating faster cAMP clearance (Supplementary Figs. 6–9 and Supplementary Table S4). Assuming the rising cAMP levels reflect a propagating wave with the slope of cAMP gradient, we estimated a mean propagation speed of $3.32 \pm 1.19$ μm/s and an apparent diffusion coefficient of $58.9 \pm 23.8$ μm²/s ($n = 3$) (Supplementary Tables S5 and S6), consistent with values from non-ratiometric measurements (Supplementary Table S2).

### Intracellular cAMP synthesis follows stimulation sites, while signal decay initiates from the rear of the cell

In some giant cells, the sites of increase and decrease in cAMP levels coincided in some cases (Supplementary Fig. 10 and Supplementary Video 4). Furthermore, in the cell repeatedly stimulated by spontaneous cAMP signals, the site of cAMP decrease gradually shifted toward the rear end of the cell over time (Supplementary Fig. 11 and Supplementary Videos 4–7), suggesting that the incoming signal started to arrive from a direction unaligned with the previously established polarity of the cell. This result supports the hypothesis that cAMP synthesis is initiated at the site of extracellular stimulation and that cells exhibit positive chemotaxis, with the decrease in cAMP levels beginning from the rear of the cell.

To further test this hypothesis, we locally applied cAMP using a glass needle and monitored the resulting intracellular cAMP dynamics (Fig. 4 and

Supplementary Video 8). Depending on the stimulus location, cAMP synthesis could initiate at either end of the cell, while the decrease consistently started from one end (Fig. 4a, b and Supplementary Video 9). The front–rear polarity of a cell was evaluated based on signal migration direction relative to the stimulus (Fig. 4 and Supplementary Videos 8–11). These results strongly suggest that while cAMP synthesis occurs at the site of receptor activation regardless of cell polarity, the reduction in cAMP levels consistently initiates from the rear region.

### Vesicle-rich rear regions may underlie spatially biased cAMP clearance

Measurements in the giant cells revealed that intracellular cAMP synthesis was spatially dependent on the site of extracellular stimulation, whereas cAMP clearance consistently initiated from the rear, independent of the stimulus location. In several fluorescence imaging datasets, the accumulation of vesicular structures was observed at the rear regions of cells (Fig. 5a–d and Supplementary Fig. 12). This suggests that rapid cytosolic cAMP clearance at the rear of the cell may involve active uptake into vesicles and/or vesicle-mediated secretion.

Electron microscopy of the giant cells confirmed the presence of large vesicles (on the order of several μm) in regions lacking filopodia, suggesting that these regions correspond to the rear side of the cell, consistent with fluorescence observations (Fig. 5e). In addition, smaller vesicular structures ranging from tens to hundreds of nm in diameter were also abundant in the same regions. While the large vesicles may be characteristic of the giant cells due to their size, the clusters of smaller vesicles appear to fall within a reasonable size range even in normal-sized cells. Previous studies have suggested that vesicle-mediated secretion from the rear region may contribute to directional cAMP release[48,49]. Our data support this model and further suggest that vesicle accumulation at the rear region may facilitate efficient spatial control of cAMP dynamics in migrating *Dictyostelium* cells.

### Biphasic and directional propagation of intracellular Ca²⁺ signals in giant *Dictyostelium* cells

In *D. discoideum*, transient increases in cytosolic $Ca^{2+}$ occur in response to cAMP signaling and activate cell motility[8,12,50]. However, due to the small sizes of typical *Dictyostelium* cells, the spatial dynamics of intracellular $Ca^{2+}$ signaling remain poorly understood. To address this and to demonstrate the versatility of signal measurements in giant cells, we expressed the genetically encoded $Ca^{2+}$ indicator GCaMP6s[14,46] in wild-type AX2 cells and induced giant cell formation to enable high-resolution analysis of intracellular $Ca^{2+}$ dynamics.

In these giant cells, we observed propagating $Ca^{2+}$ signals undetectable in normal-sized cells (Fig. 6a, b and Supplementary Video 12). Between 235 and 245 s after imaging began, cytosolic $Ca^{2+}$ levels increased in the central region of the cell, followed by a decrease in the front region between 250 and 260 s. To quantify signal directionality, we fitted the trajectory of the cell centroid to a linear function (slope = 0.19) (Fig. 6c, d). Based on this, we defined the front and rear positions relative to the centroid and analyzed the $Ca^{2+}$ signal at these locations (Fig. 6e). The resulting time courses suggested that the $Ca^{2+}$ signal exhibited a biphasic response to each cAMP pulse (Fig. 6f).

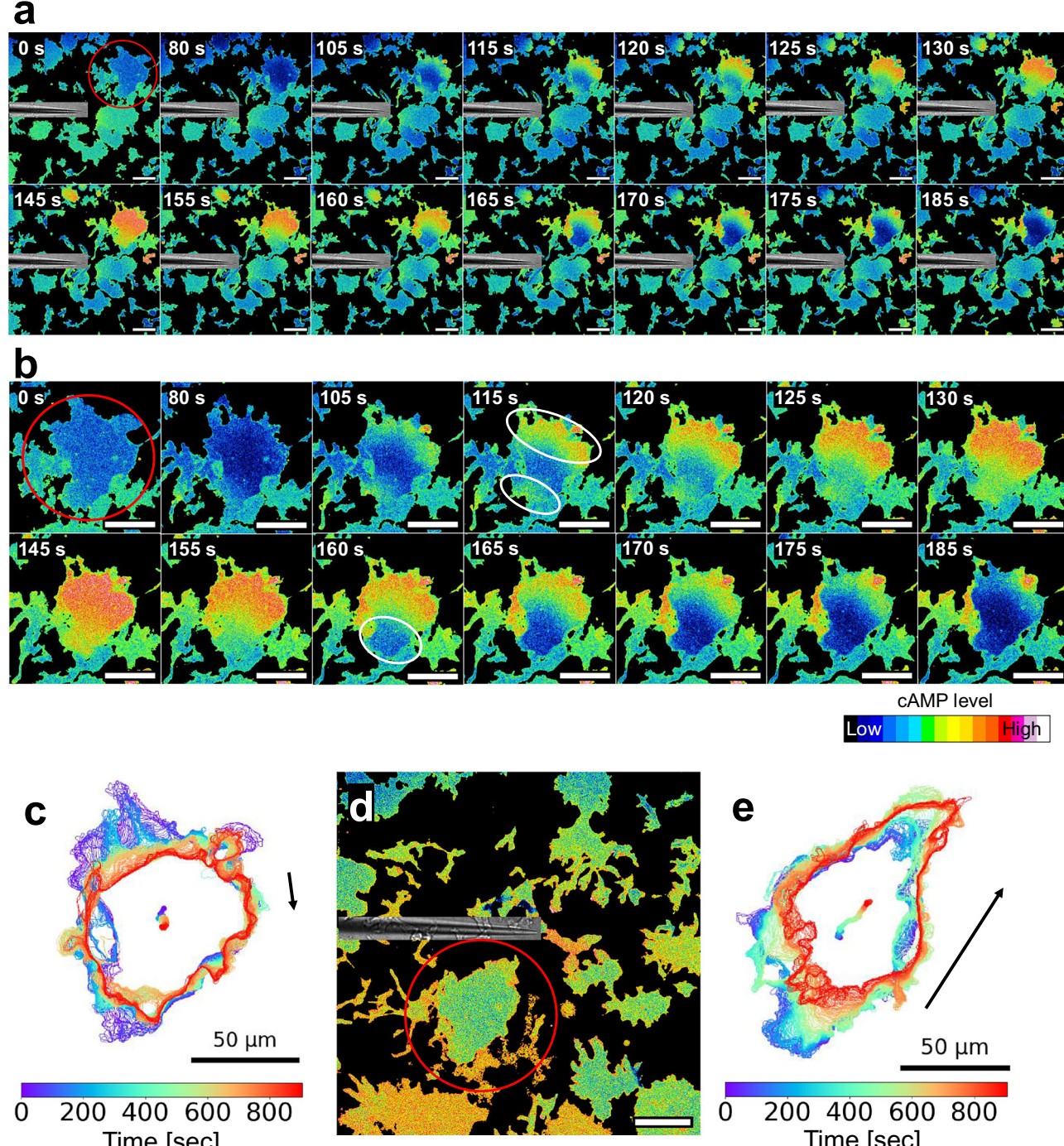

**Fig. 4 | Directional control of cAMP increase and decrease upon localized stimulation.** Giant AX2 cells expressing Flamindo2-RFP were generated by adding 10 μM blebbistatin and culturing under shaking conditions. Eight hours after starvation with DB, time-lapse fluorescence imaging was initiated using a confocal microscope. A micromanipulator-mounted glass needle was used to locally deliver 10 μM cAMP. Imaging conditions: 40× objective, 5 s intervals. **a** Ratiometric images of the entire field of view were generated by dividing the RFP signal by the Flamindo2 signal. Differential interference contrast (DIC) images of the glass needle were overlaid. Time elapsed from the start of observation is shown in the upper left corner. Scale bars: 50 μm. The red circle indicates a representative giant cell within the field. **b** Cropped images of a representative giant cell in (**a**). The white circles at 115 s indicate regions of cAMP increase; the circle at 160 s indicates a region of cAMP signal decrease. Time is indicated in the upper right corner. Scale bars: 50 μm. **c** Cell contour and centroid trajectory of the giant cell in (**b**). An arrow indicates the direction of cell movement. **d** Same as (**a**), with the micromanipulator repositioned. Scale bar: 50 μm. **e** Cell contour and centroid trajectory after repositioning of the micromanipulator. An arrow indicates the direction of cell movement.

To further investigate the spatial structure of $Ca^{2+}$ signaling, we divided each giant cell into a 7 × 7 grid and analyzed $Ca^{2+}$ transients in each grid unit. The slopes and half-times of both the first and second $Ca^{2+}$ peaks were extracted. During the first phase, only the decay slope showed clear spatial directionality, decreasing from the front to the rear. As shown in Fig. 6 and Supplementary Fig. 13, half-time analysis revealed that the first $Ca^{2+}$ transient was initiated from the upper-left region of the cell (in image coordinates) and decayed from front to rear (Fig. 6g). In contrast, the second $Ca^{2+}$

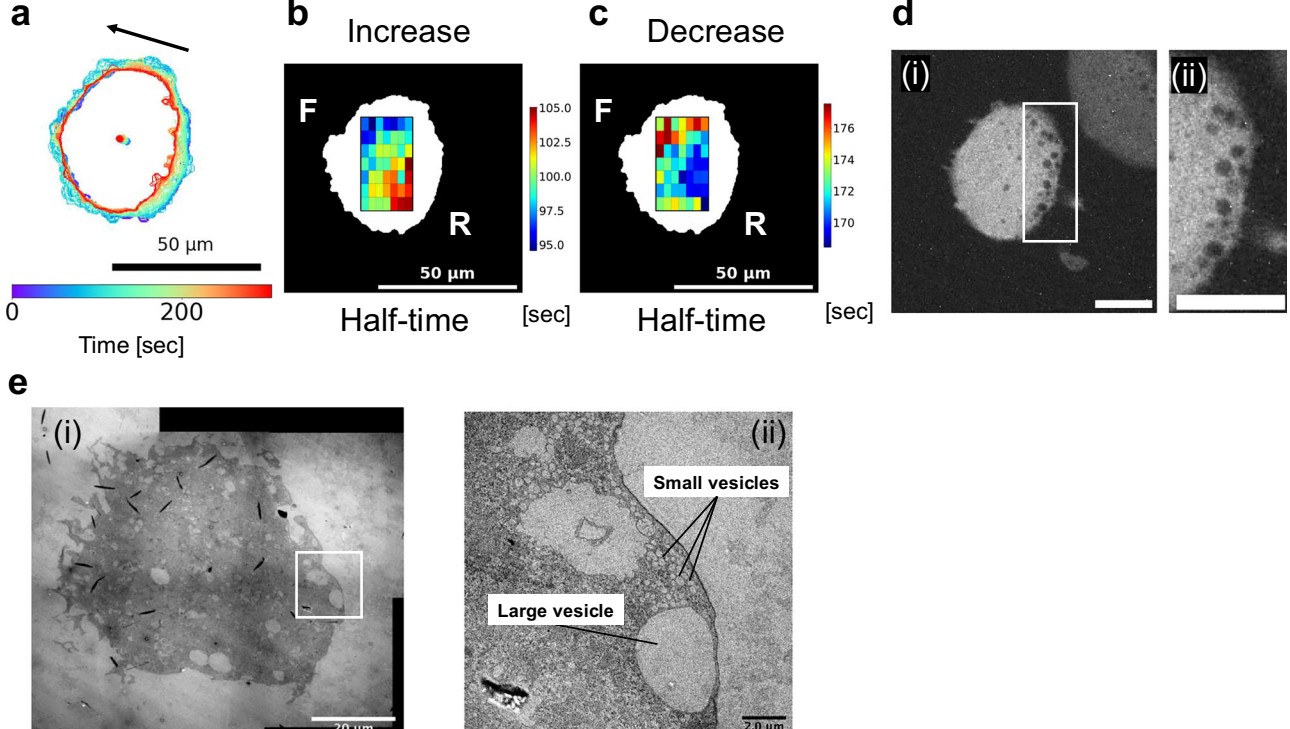

**Fig. 5 | Vesicle accumulation and cAMP decay in the rear region. a** Cell contour and centroid trajectory of a giant AX2 cell expressing Flamindo2-RFP after being starved for 7.5 h. The cell was observed under a confocal microscope at 5 s intervals. Bar: 50 μm. **b** Heatmap of the half-time parameter obtained from sigmoid fitting of the cAMP increase in each grid. **c** Heatmap of the half-time parameter obtained from sigmoid fitting of the cAMP decrease. **d** Fluorescence image showing vesicle accumulation in the rear of the cell, coinciding with the region of rapid cAMP decay in (**c**). **ii** is a magnified view of the white rectangle shown in (**i**). Scale bars: 20 μm. **e** Transmission electron microscopy (TEM) image showing both large and small vesicles concentrated in the rear region of the giant cell. **ii** is a magnified view of the white rectangle shown in (**i**). Scale bars: 20 μm in (**i**), 2 μm in (**ii**).

transient began in the rear region and propagated toward the front, with a matching decay pattern (Fig. 6h).

These results suggest that intracellular $Ca^{2+}$ signaling in *Dictyostelium* is both biphasic and directionally structured.

### Spatial coordination between $Ca^{2+}$ signaling and F-actin waves in giant cells

To explore the relationship between the directionality of $Ca^{2+}$ signals and cell motility suggested in Fig. 6, we examined the spatiotemporal dynamics of F-actin, a key driver of motility, in relation to $Ca^{2+}$ fluctuations. We introduced the F-actin-binding probe Lifeact14-mScarletI into AX2 cells expressing GCaMP6s and visualized $Ca^{2+}$ signaling alongside F-actin dynamics in the giant cells (Fig. 7, Supplementary Fig. 14, and Video 13).

As shown in Fig. 7a, $Ca^{2+}$ signaling was initiated near the center of the cell 15 s after the start of imaging and decreased progressively from the front to the rear between 20 and 35 s. During this period, F-actin localized to the cell cortex at 15 s and formed a wave-like structure that propagated from the front to the rear between 25 and 50 s (Fig. 7c). The location of the $Ca^{2+}$ signal decrease overlapped with that of the actin wave, suggesting a close spatial relationship between the two events.

Furthermore, a second phase of $Ca^{2+}$ excitation occurred at the front of the cell, concomitant with the rearward localization of the actin wave (Fig. 7e). Notably, the $Ca^{2+}$ signals and actin waves showed mutually exclusive localization. This inverse relationship was further supported by kymograph analysis (Fig. 7f).

These results suggest that the initial $Ca^{2+}$ burst triggers the propagation of the actin wave, which, in turn, corresponds with a spatial reduction in $Ca^{2+}$ levels as the actin wave traverse the cell. The second $Ca^{2+}$ burst phase appears to emerge following the relocation of the actin wave to the rear of the cell, indicating a possible feedback loop between $Ca^{2+}$ signaling and actin dynamics during chemotactic migration.

### Giant cells expand the spatiotemporal range of signal and cytoskeletal visualization beyond the limits of normal-sized cells

The use of giant cells enabled the visualization of intracellular signal propagation phenomena undetectable in normal-sized *Dictyostelium* cells. Conventional super-resolution microscopes are designed for targets with defined structures[25,26]. Therefore, we focused on F-actin, a key component in chemotactic movement, and conducted measurements by examining giant cells with super-resolution microscopy.

This approach revealed mesh-like actin networks supporting the cell cortex, as well as dynamic actin structures involved in cell–substrate interactions during migration (Supplementary Fig. 15a, b and Supplementary Videos 14 and 15). Notably, some actin-rich regions extended-over 10 μm, exceeding the typical long axis of a normal *Dictyostelium* cell (approximately 8 μm), demonstrating that giant cells allow expanded spatial analysis of cytoskeletal dynamics that are otherwise inaccessible.

In electron microscopy, although high magnification enables visualization of fine structural detail, identifying specific subcellular compartments can be challenging due to the limited field of view. Giant cells, by contrast, offer spatially expanded regions corresponding to cellular front and rear polarity, increasing the likelihood of capturing relevant structures within a single field. Indeed, electron microscopy of the large pseudopodial regions in giant cells enabled efficient visualization of the bundled actin structures responsible for protrusion formation (Supplementary Fig. 15c).

### Discussion

Understanding the spatiotemporal organization of intracellular signaling remains challenging due to the limited resolution of normal-sized cells. In this study, we established a method to generate multinucleated giant *D. discoideum* cells by partially inhibiting cytokinesis through chemical and mechanical

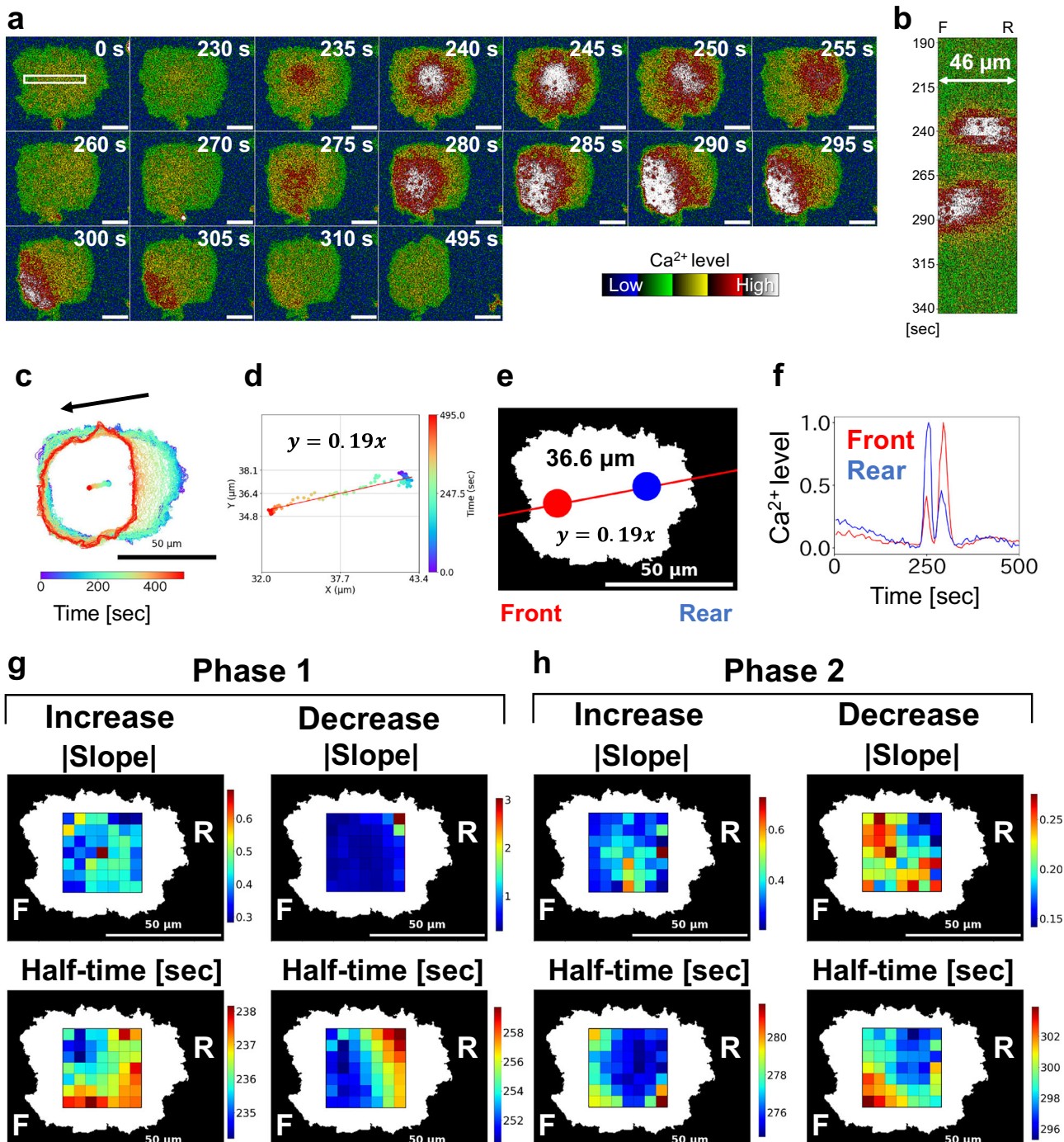

**Fig. 6 | Intracellular Ca²⁺ signaling dynamics in a giant cell.** AX2 cells expressing GCaMP6s were cultured with 10 μM blebbistatin for 6 days under shaking conditions. After starvation with DB, imaging was performed using confocal microscopy (40× objective, 5 s intervals). **a** Representative time-lapse fluorescence images of a single giant cell. Time elapsed from the beginning of imaging is shown at the top right of each panel. Scale bars: 20 μm. **b** Kymograph showing spatiotemporal Ca²⁺ dynamics along a 46 μm line in white rectangle in (**a**). Vertical axis: time [sec]. **c** Cell contour overlaid with centroid trajectory. Arrow indicates the direction of cell migration. **d** Linear approximation of the centroid trajectory. **e** Analysis regions for the front and rear were defined based on the linear fit of the centroid trajectory. The origin was set at the centroid, with front and rear regions defined at ±18 μm along the trajectory (slope = 0.19). Red and blue circles indicate front and rear regions, respectively. Region diameter: 11 μm. **f** Temporal changes in Ca²⁺ levels at the front (red) and rear (blue). Horizontal axis: time [min]; vertical axis: normalized Ca²⁺ level. **g** First-phase Ca²⁺ signal analysis: spatial grid (7 × 7 within a 35 × 36 μm rectangle) of the increase phase, heatmap of the slope parameter, and heatmap of half-time parameter derived from sigmoid fitting (left panels). Heatmaps of the decrease phase during the first Ca²⁺ wave: slope and half-time (right panels). **h** Second phase Ca²⁺ signal analysis: increase phase heatmaps, slope, and half-time (left panels); decrease phase heatmaps, slope, and half-time (right panels).

means rather than genetic manipulation. These cells retained critical physiological functions, including chemotactic signaling and front–rear polarity, enabling direct visualization of intracellular signal propagation (Figs. 2 and 3).

Previous methods for constructing giant cells, such as electrofusion, required starvation prior to cell fusion, preventing synchronized developmental analysis[31,32]. In contrast, our approach allowed real-time imaging during developmental stages without pre-treatment, and the giant cells

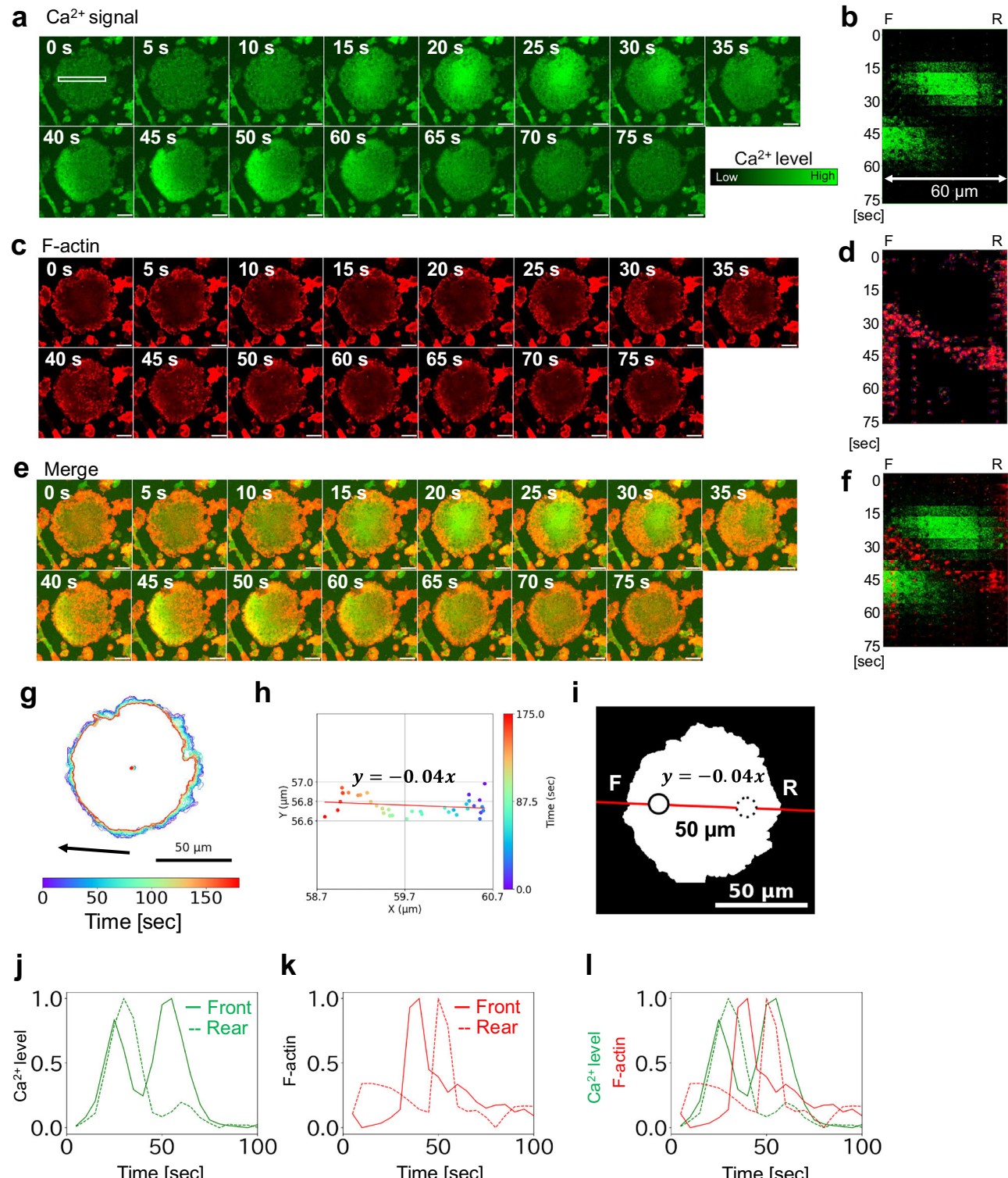

**Fig. 7 | Coordination between Ca²⁺ signals and F-actin dynamics.** AX2 cells co-expressing GCaMP6s and Lifeact14-mScarletI were cultured for 6 days under shaking conditions with 10 μM blebbistatin. After starvation with DB, imaging was performed using confocal microscopy (40× objective; 5 s intervals). **a** Representative fluorescence time-lapse images of GCaMP6s in a giant cell. Time after imaging onset is shown in the top-right corner. Scale bars: 20 μm. **b** Kymograph of Ca²⁺ signal intensity along the 60 μm region within the white rectangle indicated in (**a**). Vertical axis: time [sec]. **c** Time-lapse images of actin waves (Lifeact14-mScarletI channel) in the same cell. Scale bars: 20 μm. **d** Kymograph of actin wave dynamics within the same 60 μm region. Vertical axis: time [sec]. **e** Merged fluorescence images showing Ca²⁺ signals (GCaMP6s, green) and actin waves (Lifeact14-mScarletI, red). Scale bars: 20 μm.

**f** Merged kymograph of Ca²⁺ signals and actin waves. **g** Cell outline overlaid with centroid trajectory. An arrow indicates the direction of cell movement. **h** Linear approximation of the centroid trajectory. **i** Analysis regions for the front and rear were defined based on linear fit of the centroid trajectory. The origin was placed at the centroid, and the front and rear were defined at ±25 μm along the fit line (slope = −0.04). Solid and dashed circles indicate the front and rear regions, respectively. Region diameter: 11 μm. **j** Time series of normalized Ca²⁺ signal intensities in the front and rear regions. Horizontal axis: time [sec]; vertical axis: normalized Ca²⁺ intensity. **k** Time series of normalized actin wave intensities in the front and rear regions. Horizontal axis: time [sec]; vertical axis: normalized intensity. **l** Correlation between Ca²⁺ signaling and actin dynamics. The traces in (**j**) and (**k**) are superimposed.

remained viable for over 10 h. Unlike *mhcA paxB* knockout[37], which risks disrupting core signaling pathways, our approach preserves cellular responsiveness (Supplementary Fig. 2).

Although the combination of blebbistatin treatment and shaking culture induced the formation of giant cells, not all cells underwent hypertrophy under these conditions, resulting in increased heterogeneity of cell size (Fig. 1d). This variability allowed the direct comparison of giant and normal-sized cells within the same culture environment (Supplementary Fig. 2)[51].

Using this platform, we visualized directional cAMP synthesis propagating from the site of signal reception toward the rear of the cell (Figs. 2 and 3 and Supplementary Fig. 16). Our direct measurements revealed that both the initiation and decay of cAMP signals occur asymmetrically, with faster synthesis and sharper signal decline at the rear. These dynamics suggest a self-organizing, excitability-like process[52,53]. Notably, the apparent diffusion coefficients (40.8–85.8 $\mu m^2/s$) were slower than values from prior estimations[21] (Supplementary Table S5), likely reflecting intracellular buffering or compartmentalization by protein kinase A (PKA), phosphodiesterases (PDE), or cytoskeletal structures[54,55]. Although cAMP synthesis and decay were analyzed using sigmoid function fitting, regions with poor fitting were observed at the onset of synthesis in the rear and at the onset of decay in both the front and rear of the cell (Fig. 3d, e). These regions may involve intracellular sequential reactions that are not captured by the current simple analytical model. In the analysis areas of the giant cells, a substantial number of receptors and adenylyl cyclases contributing to cAMP increase, as well as numerous vesicles and transporters involved in cAMP decrease, are inevitably encompassed. As this heterogeneity may underlie the variability observed in the early responses, additional verification, including expression-level control or the use of relevant mutants, will likely be required.

Intracellular cAMP synthesis was initiated at the front of the cell, corresponding to the direction from which the external signal was received (Fig. 3 and Supplementary Fig. 16). The activation of adenylyl cyclase A (ACA), the primary enzyme responsible for cAMP production in the cAMP signaling relay, is regulated by CRAC. CRAC contains a PH domain that specifically localizes to a PI(3,4,5)P3-enriched region at the front of the cell, highlighting the importance of front-directed spatial control mediated by phosphoinositide signaling[53,56]. In contrast, ACA predominantly localizes at the rear of the cell. This suggests a spatial division of function, where precise CRAC-mediated activation at the front is complemented by abundant ACA at the rear of the cell[57,58]. Such organization may enable the cell to maintain front–rear polarity while ensuring a robust and coordinated cAMP response throughout the entire cell. The observation that cAMP signal initiation occurs independently of cell polarity and aligns with the direction of signal reception is consistent with the uniform distribution of the cAMP receptor cAR1 across the entire cell membrane[59]. In contrast, electron microscopy and fluorescence imaging revealed vesicle accumulation at the rear of the cell (Fig. 5), supporting the hypothesis that spatially regulated secretion underlies the abrupt signal decay observed[48].

We also uncovered a biphasic $Ca^{2+}$ signaling response (Fig. 6), which was not apparent in previous measurements and was not spatially resolved[12,14]. Calcium waves promote actin remodeling, which in turn promotes pseudopod formation[60,61]. The first $Ca^{2+}$ spike appeared temporally linked to initial cAMP synthesis, while the second occurred in association with rear-localized actin waves (Fig. 7). The first phase of the $Ca^{2+}$ signal, which is synchronized with the cAMP signal, is thought to be largely mediated by vesicle-derived $Ca^{2+}$ release via the IP3 receptor homolog IplA[62,63]. However, whether IplA also contributes to the second phase of the $Ca^{2+}$ signal or whether other pathways play a major role remains unclear. Inhibition of IplA significantly reduces the overall calcium signal[8,63], complicating the assessment of its specific involvement at this stage. Dual imaging of $Ca^{2+}$ and F-actin confirmed this temporal sequence, which was consistent with studies linking rearward $Ca^{2+}$ increases to myosin II localization[64] and transient $Ca^{2+}$ spikes in the front of the cell to actin remodeling during turning[65]. The exclusive relationship between calcium waves and actin waves may be due to the promotion of actin

depolymerization in regions of high $Ca^{2+}$ concentration[61,66]. The $Ca^{2+}$ signals mediated by IplA, such as those observed here, are not considered essential for chemotaxis to cAMP[62,67]. Consistently with this, some responses displayed clear front–rear differences, whereas others showed more spatially uniform patterns (Figs. 6 and 7). Moreover, unlike the cAMP signals, the $Ca^{2+}$ responses were not uniform and exhibited substantial variability in response levels across individual cells (Supplementary Fig. 2a, b). To clarify the relationship between $Ca^{2+}$ dynamics and chemotactic migration behavior, simultaneous imaging of spatial Ras/PI3K signaling patterns would likely be required[68].

Finally, combining the giant cell system with super-resolution and electron microscopy enabled the visualization of actin structures exceeding 10 μm, far beyond the diffraction limit or size constraints of conventional *Dictyostelium* cells (Supplementary Fig. 15). Although such structures may be exaggerated as a consequence of cell enlargement, the use of giant cells provides a valuable approach that expands the possibilities for visualizing intracellular architecture.

The giant cells established in this study were generated through suspension culture. However, upon switching to static conditions on a dish before imaging, a substantial number of cells underwent division, resulting in size reduction. In the future, elucidating the mechanisms by which giant cells maintain their large size will lead to the development of a more efficient method.

Our results provide direct evidence that intracellular chemotactic signaling proceeds via structured propagation of cAMP and $Ca^{2+}$ signals coupled with cytoskeletal feedback. This spatiotemporal organization underlies dynamic polarity establishment and directional movement. By overcoming physical constraints through cytokinesis-inhibited giant cells, we offer a versatile platform for high-resolution single-cell studies. Although not addressed in this study, enlarged cells are expected to be useful for applications such as optogenetic manipulation at the subcellular level and gene expression analysis in specific cellular regions. This strategy is applicable beyond *Dictyostelium*, with potential relevance for modeling subcellular dynamics in other systems where spatial resolution remains a limiting factor. Indeed, the spontaneous generation of giant cells has been documented in mammalian and plant systems[69,70], and suspension culture techniques have been well established for widely used model cell lines such as HeLa and iPS cells[71,72], which are typically maintained under adherent conditions. Together, these factors suggest that our enlargement approach may be applicable to these and other eukaryotic cell types.

## Materials and methods
### Cell strains and culture conditions
The *Dictyostelium discoideum* strains used in this study are listed in Supplementary Table S7. Cells were grown axenically in HL5 medium, including glucose (Formedium, Swaffham, UK) in static culture dishes (BIO-BIK I-90-20) at 21 °C. Transformants were maintained in HL5 medium supplemented with 10 μg/mL G418, 50 μg/mL hygromycin B, or 10 μg/mL blasticidin S (all from Fujifilm Wako, Osaka, Japan).

### Static culture with blebbistatin
Exponentially growing cells (up to $1 \times 10^7$ cells/mL) were diluted to $1 \times 10^5$ cells/mL and cultured in 10 mL HL5 medium containing 10 μM blebbistatin (BioGems, Westlake Village, CA, USA) at 21 °C until confluency was reached.

### Shaking culture
Cells grown to approximately $1 \times 10^7$ cells/mL under static conditions were diluted to $1 \times 10^5$ cells/mL with 10 mL HL5 medium with or without 10 μM blebbistatin in 50 mL conical flasks. The cells were cultured at 21 °C under constant rotation shaking (100 rpm) for 4–6 days.

### Plasmid construction and genetic manipulation
The plasmids used in this study are listed in Supplementary Table S8. pDM358/Dd-GCaMP6s was constructed by inserting a codon-optimized

GCaMP6s fragment[14] into the BglII and SpeI sites of pDM358[73]. pDM358/Lifeact14-mScarletI was constructed by appending a Lifeact14[74,75] sequence to the mScarletI[76] fragment via PCR using primers (Supplementary Table S9) and cloning into pDM358.

Wild-type AX2 cells were transformed by electroporation using the MicroPulser system (Bio-Rad, Hercules, CA, USA) with 1.5 µg plasmid DNA[77]. Stable transformants were selected using 50 µg/mL hygromycin B or 10 µg/mL G418.

## Cell area measurement
Cells were harvested from HL5 medium and diluted to $5 \times 10^5$ cells/mL using a cell counter (LUNA-II; Logos Biosystems, Anyang, South Korea). They were transferred to 35 mm glass-bottom dishes (3910-035; IWAKI, Tokyo, Japan) and allowed to adhere. After being washed three times with development buffer (DB: 5 mM $Na_2HPO_4$, 5 mM $KH_2PO_4$, 2 mM $MgCl_2$, 0.2 mM $CaCl_2$, pH 6.5), the cells were imaged immediately using phase-contrast microscopy (Olympus CKX53, 10× objective, NA 0.25; Olympus, Tokyo, Japan). Image segmentation was performed using Ilastik[78], and area analysis was conducted using Fiji[79].

## Signal induction by starvation
To induce signaling, cells were plated onto glass-bottom dishes and allowed to adhere. HL5 medium was replaced with DB (without blebbistatin) via three washes. Cells were starved in DB for more than 7 h at 21 °C to promote aggregation and signal relay.

## Microneedle-based cAMP stimulation
A microneedle (1 µm tip diameter; Eppendorf, Hamburg, Germany) filled with 10 µM cAMP (Sigma-Aldrich, St. Louis, MO, USA) was mounted on a micromanipulator (MM-94 and MMO-4; Narishige, Tokyo, Japan) connected to a microinjector (FemtoJet 4i; Eppendorf). Pulses were delivered at 15–20 hPa.

## Imaging and image analysis
Live-cell imaging was conducted at 21 °C using a spinning disk confocal unit (CSU-W1; Yokogawa, Tokyo, Japan) mounted on an inverted microscope (IX83; Olympus) with an sCMOS camera (Prime 95B; Teledyne Photometrics, Tucson, AZ, USA). The objectives used were the UPLXAPO 20×/0.8 NA and 40×/0.95 NA (Olympus). Fluorophores were excited using a laser diode illuminator (LDI; 89 North, Williston, VT, USA) as follows: 488 nm for Flamindo2, CRAC-GFP, and GCaMP6s; 555 nm for mCherry-Lifeact, Lifeact14-mScarletI, and mRFPmars. Super-resolution imaging was performed using the CSU-W1 SoRA module (Yokogawa) with a UPLXAPO 100×/1.45 NA objective (Olympus). All images were processed using Fiji.

## Electron microscopy
AX2 cells expressing GCaMP6s and Lifeact14-mScarletI were cultured at $1 \times 10^6$ cells in 10 mL HL5 medium with G418, hygromycin B, and 10 µM blebbistatin for 5 days under shaking (100 rpm, 21 °C). Cells were plated onto carbon-coated glass slides and incubated for 1 h at 21 °C for attachment.

Cells were fixed with 2.5% glutaraldehyde and 2% paraformaldehyde for 1 h at 4 °C, washed 3 times with KK2 buffer ($2.2 \times g$ $KH_2PO_4$, $0.7 \times g$ $K_2HPO_4/L$) on ice, and post-fixed with 1% $OsO_4$ for 1 h. After washing, dehydration was performed via a graded ethanol series (30–100%), followed by propylene oxide substitution and Spurr resin embedding (1:1 PO:Spurr, then 100% Spurr). Polymerization was conducted at 70 °C for 24 h. Ultrathin sections (40–50 nm) were prepared using a ultramicrotome (UC7; Leica, Wetzlar, Germany), double-stained with 2% uranyl acetate and lead citrate, and imaged using a transmission electron microscope (Tecnai G2 Spirit; FEI Company, Hillsboro, OR, USA) at 120 kV.

## Fluorescence signal quantification
**cAMP level estimation**. Fluorescence intensities ($I$) were processed by inverted normalization:

$$I_a = I - \max(I)$$

$$I_b = \frac{I_a}{\min(I_a)}$$

Here, $I_b$ represents the relative cAMP level. For protein concentration correction, ratiometric imaging with Flamindo2 and mRFPmars was used.

## $Ca^{2+}$ level estimation
The normalized fluorescence intensity was calculated as:

$$I_a = I - \min(I)$$

$$I_b = \frac{I_a}{\max(I_a)}$$

where $I_b$ is the relative $Ca^{2+}$ concentration.

## Sigmoid curve fitting for slope and half-time
Signal increase and decrease kinetics were modeled using a sigmoid function:

$$l = \frac{1}{1 + e^{-a(t-b)}}$$

where $L$ is the signal level, $t$ is time, $a$ is the slope, and $b$ is the half-time.

## Determination of the direction of intracellular signal propagation
Partial derivatives were calculated by forward, backward, or central difference as follows:
For x-direction:

$j = 0$: forward difference $\quad \frac{\partial T(i,j)}{\partial x} = \frac{T(i,j+1) - T(i,j)}{\Delta x}$

$j = 6$: backward difference $\quad \frac{\partial T(i,j)}{\partial x} = \frac{T(i,j) - T(i,j-1)}{\Delta x}$

otherwise: central difference $\quad \frac{\partial T(i,j)}{\partial x} = \frac{T(i,j+1) - T(i,j-1)}{2\Delta x}$

For y-direction:

$i = 0$: forward difference $\quad \frac{\partial T(i,j)}{\partial y} = \frac{T(i+1,j) - T(i,j)}{\Delta y}$

$i = 6$: backward difference $\quad \frac{\partial T(i,j)}{\partial y} = \frac{T(i,j) - T(i-1,j)}{\Delta y}$

otherwise: central difference $\quad \frac{\partial T(i,j)}{\partial y} = \frac{T(i+1,j) - T(i-1,j)}{2\Delta y}$

Gradient vector: $\nabla T(i,j) = \left( \frac{\partial T}{\partial x}(i,j), \frac{\partial T}{\partial y}(i,j) \right)$.
Gradient slope: $\frac{\bar{g}_y}{\bar{g}_x}$, which is the ratio of $\bar{g}_x = \frac{1}{N}\sum_{i,j}\frac{\partial T}{\partial x}(i,j)$ to $\bar{g}_y = \frac{1}{N}\sum_{i,j}\frac{\partial T}{\partial y}(i,j)$.
where $i$ is the row number, $j$ the column number, and $T$ the half-time parameter corresponding to $b$ in the sigmoid curve fitting.

## Cell growth and division model
We simulated changes in cell-size distribution in a growing population using a discrete-time stochastic model. The simulation started with a fixed number of identical cells $N_0$, each with an initial size $S_0$. At each time step, the simulation proceeded through two phases: growth and division.

During the growth phase, all cells synchronously doubled in size, representing exponential proliferation.

In the subsequent division phase, each cell was evaluated to determine whether it divides. Division occurred stochastically: for each cell exceeding a minimum size threshold $S_{min}$, division was triggered with a probability $P_{div}$

$\in [0.0, 1.0]$. This represents a stochastic extension of the classical sizer strategy[80], where division depends not only on size but also on chance.

If a cell divided, its size $s$ was partitioned asymmetrically into two daughter cells based on a randomly sampled ratio $p \in [p_{min}, p_{max}]$, such that:

$$s_1 = p \cdot s$$

$$s_2 = (1 - p).s$$

where $s_1$ and $s_2$ are the sizes of the daughter cells.

This process is repeated over $T$ time steps, leading to an exponentially growing population with heterogeneous cell sizes.

Notably, the total cell area in this model remained conserved as:

$$Total\ size = N_0 \times S_0 \times 2^T$$

However, the number of cells generated depends on the parameters governing the division stage. Cells that grow but fail to divide (due to division inhibition) may become multinucleated, as observed experimentally (Fig. 1). We assume that such inhibition affects only the division phase and not cell growth.

At each time step, the size distribution was recorded. The final distributions were visualized using histograms and violin plots on a logarithmic scale (as shown in Supplementary Fig. 1).

This modeling framework facilitates the investigation of how the emergent distribution of cell sizes in expanding populations results from the cumulative interplay of size-dependent thresholds, stochastic division timing, and asymmetric partitioning.

## Data availability

Data supporting the findings of this study are available from the corresponding authors upon request. The original image data acquired in this study are available at ssbd-repos-000452, https://doi.org/10.24631/ssbd.repos.2025.07.452 in SSBD:repository[81].

## Code availability

All custom code used in this study is available on GitHub at https://github.com/226E0326/cAMP_analyze.

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

## Acknowledgements

We acknowledge Yuko Baba and Chihiro Fukunaga for technical assistance. This research was supported in part by JST SPRING (Grant Number JPMJSP2154 to Y.H.) and JST PRESTO (Grant Number JPMJPR204B to Y.V.M.).

## Author contributions

Y.H. conceived the study, performed the experiments, analyzed the data, and drafted the original manuscript. Y.G. performed the experiments and analyzed the data. C.O. performed the simulation experiments and analyzed the data. T.Y. contributed to the interpretation of the results. Y.V.M. conceived and designed the study, performed the experiments, and drafted the original manuscript. All authors critically reviewed and revised the manuscript draft and approved the final version for submission.

## Competing interests

The authors declare no competing interests.
