## [Transparent Peer Review file · Communications Biology]

Establishing functional giant *Dictyostelium* cells reveals front–rear polarity in intracellular signaling

Corresponding Author: Professor Yusuke Morimoto

Version 0:

Reviewer comments:

Reviewer #1

(Remarks to the Author)

Hayashida et al., make use of giant, multinuclear *Dictyostelium* amoeba to live-image intracellular signalling, which is otherwise too rapid to spatially resolve in normal sized cells. The use of these giant cells is an ingenious approach, so much so that it is hard to believe that no one else has thought of it before. Furthermore, this approach complements recent advances in expansion microscopy and super resolution microscopy, one which is particularly well-suited for live-imaging. Firstly, they establish and verify a method to generate enlarged, multinucleate cells, one that doesn't require mutation, which might cause pleiotropic effects in any subsequent analysis. By combining these cells with biosensors and high-resolution live-imaging, the authors are successfully able to resolve intracellular imaging. Within the context of the field's extensive mechanistic understanding of cAMP signalling, this approach allows the authors to make some very detailed and powerful observations about cAMP intracellular signalling, which I consider merit publishing alone. However, given the conflicting role of Calcium signalling in *Dictyostelium* motility, I am far less convinced of this aspect of the paper. Furthermore, I am extremely dubious of the value of the use of these giant cells for live-imaging the actin cytoskeleton and these data seem little more than a flimsy add-on. I genuinely believe the latter two only serve to detract from the true value of the former and it is my recommendation that they are removed, following which I'd recommend publication with minor corrections.

Major points:

Regarding Calcium signalling:

The authors cite (but do not explicitly state the main conclusions of) Traynor et al., (2000) titled "Ca²⁺ signalling is not required for chemotaxis in *Dictyostelium*". While there is a deep, mechanistic understanding of the role of cAMP signalling in *Dictyostelium* chemotaxis, within which the observations of this paper can be rationally applied to, this is lacking regarding Calcium signalling. Therefore, without significant mechanistic interrogation of calcium's role in motility, it is difficult to conclude anything beyond correlation between the observed calcium signalling and chemotaxis. While the observed correlation between calcium waves and actin is intriguing, there could be many explanations for this other than chemotaxis (such as attempts at cytokinesis to resolve their multinuclearity, which is evident from many of the low-magnification movies).

Furthermore, the initial introductory data to calcium signalling in giant cells is weak, limited to a single, unquantified movie (movie 2), which I don't find particularly convincing. While a few giant cells very obviously flicker, many don't, nor do any normal sized ones (as far as I can tell). A detailed analysis similar to that in figures S2a-d would be required to convince me otherwise.

Regarding suitability of giant cells for studying actin dynamics:

While the authors propose that giant cell make live-imaging and EM of the cytoskeleton more convenient I am not convinced of the value of this to advancing our understanding of actin dynamics in normal cells. Firstly, while giant cells convincingly respond to cAMP waves, they are very morphologically different to normal sized chemotaxing *Dictyostelium* and are far less elongated and polarised (even if they retain biochemical polarisation). The authors highlight that giant cells allow the resolution of actin-based structures that are even larger than a whole normal cell. However, clearly a structure of this scale could never exist in a normal cell. I feel the use of such giant cells to study the cytoskeleton would be fraught with potential artifacts. For this to be included, I think a battery of quantification on giant cell motility would be required to demonstrate that

these cells are moving ~normally to support appropriate extrapolation to normal chemotaxis (i.e. speed, directionality/chemotaxis, SCAR or Arp2/3 localisation vs. actomyosin, etc).

Other major points:

While there is general localisation of the CRAC-GFP probe to the front of the both the normal and giant cells (Figure S2f & h), it appears visually very different in the latter. This is also not helped by the example of the former showing streaming cells vs. the free moving giant cell, I don't think this is a fair comparison, especially if only representative cells are being shown, without quantification.

In lines 130-131 ("Furthermore, the giant cells also demonstrated positive chemotactic migration toward cAMP indistinguishable from that of control cells"), the authors make an extremely strong statement without any quantifiable proof beyond a single qualitative movie. The authors should provide quantification of giant cell vs. normal cell chemotaxis (e.g. Forward migration index, Chemotaxis index) to justify this conclusion.

In lines 191-192 the author's admit that "Notably, such acceleration was not consistently observed across all recorded cells, indicating potential variability in the propagation (Supplementary Fig. 3)." As best I can tell, the referred to acceleration is observed in 1 cell out of a total of 6 analysed in the paper. As such, I'm not sure there this claim is justifiable, even with the above caveat.

Finally, I find the presentation of the movies isn't great, which is especially disappointing given the exquisite imaging performed. I found the vast majority played too fast, requiring me to manually move frame to frame to observe the wave like progress of the signalling.

Furthermore, movie 1 could benefit from having some overlaid tracks for multicellular cells. In movie 6, please highlight in which direction the needle is providing cAMP with (e.g.) an arrow.

Minor Points:

Figure 2: In the presented stills (a) I see no high cAMP blue like shown in kymograph (b), yet they share the same intensity key. While most of the 'blue' period shown in kymograph is not represented in the stills, I think this is potentially confusing and surely the 200s timepoint should show some blue if the intensity key is constant between the stills and the kymograph?

Please mark the front/rear directly on Fig 3 f(i-ii), g(i-ii) & in S3 f(i-ii), g(i-ii), m(i-ii) & n (i-ii) to aid interpretation of these figures, given the front rear changes in each figure.

In lines 221-227 the authors state that "In some giant cells, cases were observed in which the sites of increase and decrease in cAMP levels coincided (Supplementary Fig. 5 and Supplementary Video 5). Furthermore, in cells repeatedly stimulated by cAMP signals, the site of cAMP decrease gradually shifted toward the rear end of the cell over time (Supplementary Fig. 5i-l). This result supports the hypothesis that cAMP synthesis is initiated at the site of extracellular stimulation and that cells exhibit positive chemotaxis, with the decrease in cAMP levels beginning from the rear of the cell."

I am not sure I follow the full logic of this, perhaps because it is underexplained. How is Fig S5 consistent with the eventual hypothesis?

In lines 245-247, the authors state that "Electron microscopy of the giant cells confirmed the presence of large vesicles (on the order of several μm) at their rear regions, apparently due to the absence of filopodia, which was consistent with fluorescence observations." Again, this is under explained, how does the lack of filopodia explain the appearance of large vesicles?

Reviewer #2

(Remarks to the Author)

Hayashida et al submitted a research article titled: "Establishing functional giant Dictyostelium cells reveals front-rear polarity in intracellular signaling" to the journal *Communication Biology*. This manuscript presents a novel methodology to generate multinucleated Dictyostelium discoideum "giant cells" via partial inhibition of cytokinesis using blebbistatin and shaking culture. These enlarged cells retain chemotactic competence, cAMP relay signaling, and Ca^{2+} oscillations, enabling unprecedented high-resolution visualization of intracellular signaling dynamics. The authors report directional front-rear propagation of cAMP, vesicle-associated clearance at the rear, biphasic Ca^{2+} responses, and spatial coupling of Ca^{2+} and actin waves. The work represents a significant methodological advance and provides new insights into the spatiotemporal logic of intracellular chemotactic signaling.

This is an exciting and methodologically innovative study with strong potential impact. The work is well executed, and the conclusions are largely supported by the data. I strongly support the publication in *Communication Biology*, while the manuscript would benefit from additional mechanistic validation of the vesicle-cAMP clearance link, clearer quantification of enlargement efficiency, and improved figure presentation.

Major points

1. While the observation of rear-localized vesicles coinciding with cAMP clearance is compelling, the mechanistic link remains speculative. Perturbation of vesicle trafficking pathways (e.g., pharmacological or genetic inhibition) would strengthen this claim. Similarly, the biphasic Ca^{2+} response is described in detail, but the underlying molecular mechanism (key players, such as IplA channels, vesicle-derived release) are not dissected. A more direct test would enhance mechanistic depth.
2. The enlargement protocol produces variable outcomes, with only a subset of cells becoming giant. While the authors note this variability as advantageous for side-by-side comparison, the reproducibility of the approach may be a concern for others attempting to adopt it. Quantification of the percentage of cells that successfully reach the “giant” state would be helpful.
3. The use of sigmoid fitting to extract propagation velocities and diffusion coefficients is appropriate, but some fitting failures at signal onset (noted by the authors) suggest the need for more complex models. Clarification of these limitations in the discussion is appreciated but could be expanded.
4. The manuscript would benefit from a more explicit discussion of how this approach could be generalized to other cell systems. While *Dictyostelium* is a powerful model, outlining the feasibility (or limitations) of applying similar enlargement strategies to mammalian cells would broaden the article’s appeal.

Minor points

1. Figures are generally clear, but several supplementary figures (e.g., Supplementary Fig. 5 on repeated stimulation) are central to the mechanistic argument and could be moved into the main text.
2. The terminology “propagation velocity” versus “diffusion coefficient” could be clarified to avoid confusion, as both are used to describe cAMP dynamics.
3. Some sections of the Results (e.g., grid-based mapping analyses) are highly technical. A schematic summarizing the observed propagation patterns (front synthesis, rear decay, vesicle involvement) would help readers synthesize findings.
4. The introduction could better frame why *Dictyostelium* serves as an ideal testbed for such enlargement approaches compared to other model systems.

Reviewer #3

(Remarks to the Author)

To better understand intracellular signaling dynamics in single cells, Hayashia et al. developed a method to generate giant multinucleated *Dictyostelium* cells by inhibiting contractile ring formation with a myosin II ATPase inhibitor in combination with disrupting adhesion through shaking. This approach overcame the size limitation of normal cells and enabled detailed mapping of spatiotemporal signaling dynamics in single cells. The enlarged cells retained normal signaling, polarity, and migration, allowing visualization of wave-like, asymmetric cAMP propagation aligned with polarity. They also observed that the rear ends of these enlarged cells are vesicles rich, suggesting that the rapid directional cAMP clearance at the rear is mediated by vesicle secretion. Using these enlarged cells the authors were able to capture biphasic and directional Ca^{2+} dynamics in response to cAMPs signalling. Finally, they showed that Ca^{2+} signals are anticorrelated with actin waves, a key regulator of chemotaxis. These findings highlight how enlarged cells reveal spatial organization of intracellular signaling that remains limited in normal-sized cells. Overall, I think the method and these initial characterizations are interesting and I support its publication, pending minor revision based on the following points.

Major Comments

Timing of stimulation vs. imaging

The definition of the 0 sec time point is unclear. Were cells already stimulated before imaging began? Consistent timing is critical for kinetic comparisons.

Figure 3: cAMP gradients

The authors observed heterogeneity in the intracellular spatial dynamics of cAMP, supporting wave-like propagation. They further suggested that the heatmaps derived from each grid revealed that the direction of initial cAMP synthesis matched that of subsequent cell migration. While interesting, the spatial gradients in Figure 3f appear to align more with the left-to-right trajectory in Figure 3a than with the top-to-bottom trajectory in Figure 3h. Assuming that the heatmaps in Figure 3f-g are derived from a single cAMP pulse during the 0-260sec time frame, it will be useful to show how the heatmaps change over time as the cells continue to migrate, as in Figure 3h? This is with reference to supplement movie 4 where there appears to be multiple more cAMP pulses after the 0-260sec time frame. If one were to look at the movie more closely, the direction of the spatial gradient does become more similar to the subsequent trajectory in Figure 3h.

Line 221 / Supplementary Fig. 5

“In some giant cells, cases were observed in which the sites of increase and decrease in cAMP levels coincided (Supplementary Fig. 5 and Supplementary Video 5).” This statement is unclear and requires further explanation. Nevertheless, the kinetics of cAMP increase and decrease indeed differ from the other cells showed in this work. What could be potentially different in these cells? Were these cells imaged at the same post-stimulation interval as the other cells in this paper?

Figure 4b:

It will be helpful to indicate front and rear in Fig. 3b. How does the heatmap and cAMP kinetics look like in front and rear of cell in (c) and (d). This will give a clearer understanding of how cAMP signalling is spatiotemporally regulated with respect to the site of extracellular stimulation

Figure 5:

Is it possible to show a control cell of normal size? In addition, cells may induce larger vesicles (several μm in size) due to cellular stress, independent of their sizes. The authors should comment on this possibility. In addition, there is no statistics of this phenotype.

Figure 6: Ca^{2+} coupling

The biphasic Ca^{2+} response to cAMP pulses is intriguing but requires clarification of how cAMP and Ca^{2+} signals are coupled inside of the cell.

Figure 7: Ca^{2+} -actin relationship

To be able to capture intracellular Ca^{2+} spatial dynamics and to couple it with F-actin waves is interesting and indeed limited in normal sized cells. There are however better ways to illustrate their point. It is not immediately clear how the dynamics of Ca^{2+} and F-actin are coordinated. The kymograph did not show a clear overlap. Figure 7d is barely visible. It will be helpful to overlay the time series of normalized Ca^{2+} and F-actin signal intensities from Fig I and J (but split front and back in two figures) to better understand how they are coupled with each other.

Minor Comments

- Scale bar is missing on Figure 3a, 4c, 4e, 5a, 6c, 7g.
- In the lookup table for Fig2, Fig 4, red is low, blue is high; in Fig 6a, red is high, blue is low. Please make it consistent. The conventional view is that warm color is high.
- Figure 4 legend Line 797/Line 801: time is shown in the upper left, not right.
- Figure 6b, 7b,d,f: indicate front and rear on kymographs.

Reviewer #4

(Remarks to the Author)

I co-reviewed this manuscript with one of the reviewers who provided the listed reports. This is part of the Communications Biology initiative to facilitate training in peer review and to provide appropriate recognition for Early Career Researchers who co-review manuscripts.

Version 1:

Reviewer comments:

Reviewer #1

(Remarks to the Author)

I have evaluated the corrected manuscript and I am happy that the author's have addressed my concerns in the majority of cases. The presentation and speed of the movies in particular is a big improvement. I only have 2 outstanding issues, one major, one minor:

Major point (relating to my original comment regarding "acceleration phenomenon"):

Although we examined more cell datasets than those shown in the manuscript, this particular case is relatively uncommon, though similar patterns can occasionally be detected. Therefore, we softened the wording as follows:

"Notably, such acceleration was not consistently observed across all recorded cells, implying that propagation may vary to some extent (Supplementary Fig. 4, 5)."

I still fundamentally disagree with this and find the authors' response inadequate. From the data presented, 1 cell out of 6 cells exhibit "acceleration". This implies the majority of the time there isn't acceleration and that any pattern could be (or even should be considered) random. So, I do not believe anything can be concluded or proposed based on these data and any claim or discussion about acceleration should be removed. To put it bluntly, you can't just pick one cell you saw something interesting with and try and make a claim out of it and say that all the other cells represent "some" variability. Stating that you haven't presented all analysed data isn't reassuring either. If the authors want to claim there is something to this (even softened as is), I'd argue they need to present data proving it is more common than not for it to be considered a "pattern" or, at the very least a minimum of $n=3$ observations to be considered a consistent feature.

Minor point:

"Consistently with this, we observed both responses with clear front-rear polarity and responses without such polarity (Fig. 6, 7)."

I don't find this newly added sentence very clear.

I feel very strongly about the first point. Unless I have misunderstood the data presented or the authors' can provide further examples, I do not believe this can be included in the paper as is.

Reviewer #2

(Remarks to the Author)

The authors have addressed all my concerns in the revised manuscript. I recommend the manuscript for publication without further revision.

Reviewer #3

(Remarks to the Author)

The authors have addressed all my comments with new figures or text editing. I support its publication.

Reviewer #4

(Remarks to the Author)

I co-reviewed this manuscript with one of the reviewers who provided the listed reports. This is part of the Communications Biology initiative to facilitate training in peer review and to provide appropriate recognition for Early Career Researchers who co-review manuscripts.

Responses to reviewers

We sincerely thank all three reviewers for their constructive and insightful comments. We have carefully revised the manuscript to address the concerns raised. Below we provide a detailed point-by-point response.

Reviewers' comments are in blue below, with our responses in black.

Reviewer #1:

Hayashida et al., make use of giant, multinuclear *Dictyostelium* amoeba to live-image intracellular signalling, which is otherwise too rapid to spatially resolve in normal sized cells. The use of these giant cells is an ingenious approach, so much so that it is hard to believe that no one else has thought of it before. Furthermore, this approach complements recent advances in expansion microscopy and super resolution microscopy, one which is particularly well-suited for live-imaging. Firstly, they establish and verify a method to generate enlarged, multinucleate cells, one that doesn't require mutation, which might cause pleiotropic effects in any subsequent analysis. By combining these cells with biosensors and high-resolution live-imaging, the authors are successfully able to resolve intracellular imaging. Within the context of the field's extensive mechanistic understanding of cAMP signalling, this approach allows the authors to make some very detailed and powerful observations about cAMP intracellular signalling, which I consider merit publishing alone. However, given the conflicting role of Calcium signalling in *Dictyostelium* motility, I am far less convinced of this aspect of the paper. Furthermore, I am extremely dubious of the value of the use of these giant cells for live-imaging the actin cytoskeleton and these data seem little more than a flimsy add-on. I genuinely believe the latter two only serve to detract from the true value of the former and it is my recommendation that they are removed, following which I'd recommend publication with minor corrections.

We greatly appreciate the reviewer's recognition of the novelty of our approach using giant *Dictyostelium* cells and the value of our cAMP imaging. We also thank the reviewer for highlighting concerns regarding the Ca²⁺ signaling and actin cytoskeleton analyses. We have carefully considered these points and revised the manuscript accordingly.

Major points:

Regarding Calcium signalling:

The authors cite (but do not explicitly state the main conclusions of) Traynor et al., (2000) titled "Ca²⁺ signalling is not required for chemotaxis in *Dictyostelium*". While there is a deep, mechanistic

understanding of the role of cAMP signalling in Dictyostelium chemotaxis, within which the observations of this paper can be rationally applied to, this is lacking regarding Calcium signalling. Therefore, without significant mechanistic interrogation of calcium's role in motility, it is difficult to conclude anything beyond correlation between the observed calcium signalling and chemotaxis. While the observed correlation between calcium waves and actin is intriguing, there could be many explanations for this other than chemotaxis (such as attempts at cytokinesis to resolve their multinuclearity, which is evident from many of the low-magnification movies). Furthermore, the initial introductory data to calcium signalling in giant cells is weak, limited to a single, unquantified movie (movie 2), which I don't find particularly convincing. While a few giant cells very obviously flicker, many don't, nor do any normal sized ones (as far as I can tell). A detailed analysis similar to that in figures S2a-d would be required to convince me otherwise.

We agree with the reviewer that our initial presentation of Ca^{2+} dynamics was limited. In the revised manuscript, similarly to the cAMP signal analysis, we compared intercellular signal propagation between adjacent normal-sized and giant cells (new Fig. S2b). Unlike in cAMP signaling (Fig. S2a), both normal-sized and giant cells exhibited variable activation patterns, with firing levels sometimes high and sometimes low, indicating that the response was not as uniform as that of cAMP. This variability is partly due to differences in the basal concentration levels among individual cells, which, when deviating from the probe's K_d , can lead to variations in detection sensitivity.

We cited Traynor et al. (2000) and expanded the Discussion accordingly.

The Ca^{2+} signals mediated by IplA, such as those observed here, are not considered essential for chemotaxis to cAMP^{62,67}. Consistently with this, we observed both responses with clear front-rear polarity and responses without such polarity (Fig. 6, 7). Moreover, unlike the cAMP signals, the Ca^{2+} responses were not uniform and exhibited substantial variability in response levels across individual cells (Supplementary Fig. 2a, b). To clarify the relationship between Ca^{2+} dynamics and chemotactic migration behavior, simultaneous imaging of spatial Ras/PI3K signaling patterns would likely be required⁶⁸.

In addition, we carefully performed directional analyses of the calcium signals in the same manner as for the cAMP signals (New Fig. 7, Fig. S13) and added additional examples from other cells (new Fig. S14). Correspondingly, we also reanalyzed the directional dynamics of the cAMP signals in more detail (new Fig. 3h, Fig. S3–S11), specifically comparing the directions of cell movement, cAMP signal increase, and decrease. Furthermore, we enhanced the precision of slope estimation by calculating values with sub-pixel (decimal) resolution rather than at the pixel level.

Furthermore, the present calcium imaging primarily serves to demonstrate the versatility of our giant-cell approach by providing an example of signaling other than that of cAMP. We realize that this point may have appeared underemphasized in the original version. Therefore, we added the following sentences to clarify this aspect.

To address this **and to demonstrate the versatility of signal measurements in giant cells**, we expressed the genetically encoded Ca²⁺ indicator GCaMP6s in wild-type AX2 cells and induced giant cell formation to enable high-resolution analysis of intracellular Ca²⁺ dynamics.

Regarding suitability of giant cells for studying actin dynamics:

While the authors propose that giant cell make live-imaging and EM of the cytoskeleton more convenient I am not convinced of the value of this to advancing our understanding of actin dynamics in normal cells. Firstly, while giant cells convincingly respond to cAMP waves, they are very morphologically different to normal sized chemotaxing Dictyostelium and are far less elongated and polarised (even if they retain biochemical polarisation). The authors highlight that giant cells allow the resolution of actin-based structures that are even larger than a whole normal cell. However, clearly a structure of this scale could never exist in a normal cell. I feel the use of such giant cells to study the cytoskeleton would be fraught with potential artifacts. For this to be included, I think a battery of quantification on giant cell motility would be required to demonstrate that these cells are moving ~normally to support appropriate extrapolation to normal chemotaxis (i.e. speed, directionality/chemotaxis, SCAR or Arp2/3 localisation vs. actomyosin, etc).

We recognize the reviewer's concern. To moderate the strength of our claims, we removed the statement regarding cytoskeletal imaging from the abstract. Moreover, to better evaluate cellular motility, we included a comparison between movement direction and the directionality of each signal in all analyses (new Fig. S3–S11).

As mentioned earlier regarding the calcium signals, our main intention here is to provide an example of observations obtained using the giant-cell approach. To make this clearer, we revised the relevant part of the Discussion as follows:

Although such structures may be exaggerated as a consequence of cell enlargement, the use of giant cells provides a valuable approach that expands the possibilities for visualizing intracellular architecture.

Other major points:

While there is general localisation of the CRAC-GFP probe to the front of the both the normal and giant cells (Figure S2f & h), it appears visually very different in the latter. This is also not helped by the example of the former showing streaming cells vs. the free moving giant cell, I don't think this is a fair comparison, especially if only representative cells are being shown, without quantification.

As you pointed out, although CRAC-GFP shows clear anterior localization, it sometimes appears somewhat granular. Because we were concerned that this might be a probe-specific artifact, we performed a comparison using PH-RFP and replaced those data (new Fig. S2c–e). In line with your

suggestion, we selected single, isolated normal-sized cells for comparison. We also added a time-series analysis. Although PH-RFP occasionally exhibits a punctate pattern, it is distributed toward the leading edge over time. From these results, we consider that anterior localization of PIP3 is preserved. Recent studies have suggested that the membrane domain size of Ras, which controls PIP3 production, is determined by RasGEFs (Iwamoto et al., Nat Commun 16: 117. 2025). Given that the size of the PIP3 domain appears to be intrinsically limited, the inability to form larger domains may result in the granular pattern observed in giant cells. While this phenomenon could potentially reveal a new aspect of PIP3 regulation, we believe that a detailed discussion of this point is beyond the scope of the present manuscript.

In lines 130-131 (“Furthermore, the giant cells also demonstrated positive chemotactic migration toward cAMP indistinguishable from that of control cells”), the authors make an extremely strong statement without any quantifiable proof beyond a single qualitative movie. The authors should provide quantification of giant cell vs. normal cell chemotaxis (e.g. Forward migration index, Chemotaxis index) to justify this conclusion.

Since it was not appropriate to state that the motility of the giant cells was exactly the same as that of normal-sized cells, we modified the phrasing accordingly. In addition, as mentioned earlier, we incorporated comparisons between movement direction and signal directionality into each analysis to more accurately assess cellular motility (New Fig. S3–S11).

Furthermore, the giant cells also exhibited positive chemotactic migration toward cAMP, which appeared broadly similar to that of control cells

In lines 191-192 the author’s admit that “Notably, such acceleration was not consistently observed across all recorded cells, indicating potential variability in the propagation (Supplementary Fig. 3).” As best I can tell, the referred to acceleration is observed in 1 cell out of a total of 6 analysed in the paper. As such, I’m not sure there this claim is justifiable, even with the above caveat.

Although we examined more cell datasets than those shown in the manuscript, this particular case is relatively uncommon, though similar patterns can occasionally be detected. Therefore, we softened the wording as follows:

Notably, such acceleration was not consistently observed across all recorded cells, implying that propagation may vary to some extent (Supplementary Fig. 4, 5).

Finally, I find the presentation of the movies isn’t great, which is especially disappointing given the exquisite imaging performed. I found the vast majority played too fast, requiring me to manually

move frame to frame to observe the wave like progress of the signalling.

As suggested, we adjusted the playback speed for all movies, reducing the speed to enhance clarity and ease of observation.

Furthermore, movie 1 could benefit from having some overlaid tracks for multicellular cells. In movie 6, please highlight in which direction the needle is providing cAMP with (e.g.) an arrow.

Since the addition of tracking traces reduced the visibility of the signals, we modified the movies so that the temporal changes in cell contours are shown alongside each signal video. In the pipette assay movies, we further included a marker indicating the position of the cAMP-loaded needle tip.

Minor Points:

Figure 2: In the presented stills (a) I see no high cAMP blue like shown in kymograph (b), yet they share the same intensity key. While most of the 'blue' period shown in kymograph is not represented in the stills, I think this is potentially confusing and surely the 200s timepoint should show some blue if the intensity key is constant between the stills and the kymograph?

The Figure 2 intensity scales and kymographs have been corrected for consistency.

Please mark the front/rear directly on Fig 3 f(i-ii), g(i-ii) & in S3 f(i-ii), g(i-ii), m(i-ii) & n (i-ii) to aid interpretation of these figures, given the front rear changes in each figure.

Front/rear markers have been added in the indicated figures

In lines 221-227 the authors state that "In some giant cells, cases were observed in which the sites of increase and decrease in cAMP levels coincided (Supplementary Fig. 5 and Supplementary Video 5). Furthermore, in cells repeatedly stimulated by cAMP signals, the site of cAMP decrease gradually shifted toward the rear end of the cell over time (Supplementary Fig. 5i-l). This result supports the hypothesis that cAMP synthesis is initiated at the site of extracellular stimulation and that cells exhibit positive chemotaxis, with the decrease in cAMP levels beginning from the rear of the cell."

I am not sure I follow the full logic of this, perhaps because it is underexplained. How is Fig S5 consistent with the eventual hypothesis?

To clarify the discussion, we divided the original figure into two separate panels (new Fig. S10, 11).

First, new Fig. S10 presents a rare case in which the increase and decrease of the cAMP signal occur in the same direction. We explained the underlying reason for this phenomenon using the pipette assay (Fig. 4): specifically, we interpret this as the cell responding to a signal arriving from a direction different from the one defined by its established polarity. In line with this interpretation, new Fig. S11 illustrates how the cell gradually aligns its polarity with the incoming signal from another direction. We revised the corresponding text to make this explanation clearer, as follows:

Furthermore, in the cell repeatedly stimulated by spontaneous cAMP signals, the site of cAMP decrease gradually shifted toward the rear end of the cell over time (Supplementary Fig. 11 and Supplementary Video 4–7), suggesting that the incoming signal started to arrive from a direction unaligned with the previously established polarity of the cell.

In lines 245-247, the authors state that “Electron microscopy of the giant cells confirmed the presence of large vesicles (on the order of several μm) at their rear regions, apparently due to the absence of filopodia, which was consistent with fluorescence observations.” Again, this is under explained, how does the lack of filopodia explain the appearance of large vesicles?

Because the original description was insufficient, we have revised the text as follows:

Electron microscopy of the giant cells confirmed the presence of large vesicles (on the order of several μm) in regions lacking filopodia, suggesting that these regions correspond to the rear side of the cell, consistent with fluorescence observations (Fig. 5e).

Reviewer #2:

Hayashida et al submitted a research article titled: “Establishing functional giant Dictyostelium cells reveals front–rear polarity in intracellular signaling” to the journal *Communication Biology*. This manuscript presents a novel methodology to generate multinucleated Dictyostelium discoideum “giant cells” via partial inhibition of cytokinesis using blebbistatin and shaking culture. These enlarged cells retain chemotactic competence, cAMP relay signaling, and Ca²⁺ oscillations, enabling unprecedented high-resolution visualization of intracellular signaling dynamics. The authors report directional front–rear propagation of cAMP, vesicle-associated clearance at the rear, biphasic Ca²⁺ responses, and spatial coupling of Ca²⁺ and actin waves. The work represents a significant methodological advance and provides new insights into the spatiotemporal logic of intracellular chemotactic signaling.

This is an exciting and methodologically innovative study with strong potential impact. The work is well executed, and the conclusions are largely supported by the data. I strongly support the publication in *Communication Biology*, while the manuscript would benefit from additional mechanistic validation of the vesicle–cAMP clearance link, clearer quantification of enlargement efficiency, and improved figure presentation.

We thank the reviewer for the strong support and for constructive suggestions that strengthen the mechanistic and methodological depth of our work.

Major points

1. While the observation of rear-localized vesicles coinciding with cAMP clearance is compelling, the mechanistic link remains speculative. Perturbation of vesicle trafficking pathways (e.g., pharmacological or genetic inhibition) would strengthen this claim. Similarly, the biphasic Ca²⁺ response is described in detail, but the underlying molecular mechanism (key players, such as IplA channels, vesicle-derived release) are not dissected. A more direct test would enhance mechanistic depth.

We agree with the reviewer’s point; however, we have unpublished data indicating that the susceptibility to starvation—namely, the presence or absence of cAMP signaling—is linked to the efficiency of giant-cell formation, and we are preparing a separate manuscript on this topic. Because inhibition of vesicle formation reduces giant-cell formation efficiency, it makes the measurements difficult to perform in giant cells. We therefore prefer to address this issue in the separate manuscript. In addition, regarding IplA knockout, we have added the following discussion to the manuscript to clarify the current limitations of our analysis.

The Ca²⁺ signals mediated by IplA, such as those observed here, are not considered essential for chemotaxis to

cAMP^{62,67}. Consistently with this, we observed both responses with clear front–rear polarity and responses without such polarity (Fig. 6, 7). Moreover, unlike the cAMP signals, the Ca²⁺ responses were not uniform and exhibited substantial variability in response levels across individual cells (Supplementary Fig. 2a, b). To clarify the relationship between Ca²⁺ dynamics and chemotactic migration behavior, simultaneous imaging of spatial Ras/PI3K signaling patterns would likely be required⁶⁸.

2. The enlargement protocol produces variable outcomes, with only a subset of cells becoming giant. While the authors note this variability as advantageous for side-by-side comparison, the reproducibility of the approach may be a concern for others attempting to adopt it. Quantification of the percentage of cells that successfully reach the “giant” state would be helpful.

We have incorporated the following description into the main text to clarify the proportion of giant cells observed in Fig. 1.

Whereas normally cultured cells exhibited a median area of 86.5 μm^2 , only 0.5% of cells under these conditions exceeded 200 μm^2 , a size more than twice the median (Fig. 1a). In contrast, under the conditions used in Fig. 1d, 59.5% of cells were larger than 200 μm^2 , and 20.5% exceeded 500 μm^2 , indicating that giant-cell formation was achieved with markedly higher efficiency.

3. The use of sigmoid fitting to extract propagation velocities and diffusion coefficients is appropriate, but some fitting failures at signal onset (noted by the authors) suggest the need for more complex models. Clarification of these limitations in the discussion is appreciated but could be expanded.

We have expanded the Discussion to clarify fitting limitations and the potential need for more complex models in future studies.

In the analysis areas of the giant cells, a substantial number of receptors and adenylyl cyclases contributing to cAMP increase, as well as numerous vesicles and transporters involved in cAMP decrease, are inevitably encompassed. As this heterogeneity may underlie the variability observed in the early responses, additional verification, including expression-level control or the use of relevant mutants, will likely be required.

4. The manuscript would benefit from a more explicit discussion of how this approach could be generalized to other cell systems. While Dictyostelium is a powerful model, outlining the feasibility (or limitations) of applying similar enlargement strategies to mammalian cells would broaden the article’s appeal.

We have added new statements to the Discussion addressing the potential applications of the enlargement approach in mammalian and other eukaryotic cells.

Indeed, the spontaneous generation of giant cells has been documented in mammalian and plant systems^{69,70}, and suspension-culture techniques have been well established for widely used model cell lines such as HeLa and iPS cells^{71,72}, which are typically maintained under adherent conditions. Together, these factors suggest that our enlargement approach may be applicable to these and other eukaryotic cell types.

Minor points

1. Figures are generally clear, but several supplementary figures (e.g., Supplementary Fig. 5 on repeated stimulation) are central to the mechanistic argument and could be moved into the main text.

Since the original Fig. S5 corresponds to a case that is relatively uncommon in our measurements, we have placed it in the Supplementary Figures to prevent possible misinterpretation. For greater clarity, we divided the figure into two parts (new Fig. S10, 11) and added explanatory descriptions to the main text. The correctness of this interpretation is supported by the pipette assay results shown in Fig. 4.

Furthermore, in the cell repeatedly stimulated by spontaneous cAMP signals, the site of cAMP decrease gradually shifted toward the rear end of the cell over time (Supplementary Fig. 11 and Supplementary Video 4–7), suggesting that the incoming signal started to arrive from a direction unaligned with the previously established polarity of the cell.

2. The terminology “propagation velocity” versus “diffusion coefficient” could be clarified to avoid confusion, as both are used to describe cAMP dynamics.

“Propagation velocity” refers to a one-dimensional measure of speed, whereas the “diffusion coefficient” represents a two-dimensional diffusion parameter that is commonly used as a general index of diffusivity. To improve clarity, we have revised the text accordingly as follows:

we estimated the average one-dimensional propagation velocity to be $3.46 \pm 0.22 \mu\text{m/s}$ and the apparent diffusion coefficient to be $74.6 \pm 7.0 \mu\text{m}^2/\text{s}$

3. Some sections of the Results (e.g., grid-based mapping analyses) are highly technical. A schematic summarizing the observed propagation patterns (front synthesis, rear decay, vesicle involvement) would help readers synthesize findings.

A schematic summarizing cAMP propagation (front synthesis, rear decay, vesicle involvement) has been added (new Fig. S16).

4. The introduction could better frame why Dictyostelium serves as an ideal testbed for such enlargement approaches compared to other model systems.

We expanded the Introduction to emphasize why *Dictyostelium* is an ideal model for this approach as below:

Given the well-developed genetic tools and databases available for *Dictyostelium*^{19,20}, single-cell signal measurement in this organism is a crucial yet technically demanding task due to its relatively small cell size.

Reviewer #3:

To better understand intracellular signaling dynamics in single cells, Hayashia et al. developed a method to generate giant multinucleated Dictyostelium cells by inhibiting contractile ring formation with a myosin II ATPase inhibitor in combination with disrupting adhesion through shaking. This approach overcame the size limitation of normal cells and enabled detailed mapping of spatiotemporal signaling dynamics in single cells. The enlarged cells retained normal signaling, polarity, and migration, allowing visualization of wave-like, asymmetric cAMP propagation aligned with polarity. They also observed that the rear ends of these enlarged cells are vesicles rich, suggesting that the rapid directional cAMP clearance at the rear is mediated by vesicle secretion. Using these enlarged cells the authors were able to capture biphasic and directional Ca^{2+} dynamics in response to cAMPs signalling. Finally, they showed that Ca^{2+} signals are anticorrelated with actin waves, a key regulator of chemotaxis. These findings highlight how enlarged cells reveal spatial organization of intracellular signaling that remains limited in normal-sized cells. Overall, I think the method and these initial characterizations are interesting and I support its publication, pending minor revision based on the following points.

We thank the reviewer for the supportive evaluation and for the helpful clarifications requested.

Major Comments

Timing of stimulation vs. imaging

The definition of the 0 sec time point is unclear. Were cells already stimulated before imaging began? Consistent timing is critical for kinetic comparisons.

Because our analyses focus on spontaneous signal responses, we define time 0 as either the onset of imaging or the first frame used in the analysis. In contrast, because the stimulation in Fig. 4 is applied using a micropipette, the onset of stimulation is designated as 0 sec in the presentation. To remove any ambiguity, we have revised the figures and their legends to clearly state these definitions wherever relevant.

Figure 3: cAMP gradients

The authors observed heterogeneity in the intracellular spatial dynamics of cAMP, supporting wave-like propagation. They further suggested that the heatmaps derived from each grid revealed that the direction of initial cAMP synthesis matched that of subsequent cell migration. While interesting, the spatial gradients in Figure 3f appear to align more with the left-to-right trajectory in Figure 3a than with the top-to-bottom trajectory in Figure 3h. Assuming that the heatmaps in Figure 3f-g are derived from a single cAMP pulse during the 0-260sec time frame, it will be useful to show how the heatmaps

change over time as the cells continue to migrate, as in Figure 3h? This is with reference to supplement movie 4 where there appears to be multiple more cAMP pulses after the 0-260sec time frame. If one were to look at the movie more closely, the direction of the spatial gradient does become more similar to the subsequent trajectory in Figure 3h.

In light of the reviewer's comment, we carefully re-examined our analyses. Because our previous analysis was performed at the pixel level, we switched to a sub-pixel resolution for slope estimation. With this refinement in Fig. 3, we found that the early cell movement and the cAMP signal gradient were almost identical (new Fig. 3i). Accordingly, we replaced the original panel with a new comparative figure that clearly illustrates this relationship (new Fig. 3). We also re-evaluated all other signal-gradient analyses and, for each case, added a comparison between the direction of cell movement and the direction of the signal gradient (new Fig. S3–11).

Line 221 / Supplementary Fig. 5

"In some giant cells, cases were observed in which the sites of increase and decrease in cAMP levels coincided (Supplementary Fig. 5 and Supplementary Video 5)." This statement is unclear and requires further explanation. Nevertheless, the kinetics of cAMP increase and decrease indeed differ from the other cells showed in this work. What could be potentially different in these cells? Were these cells imaged at the same post-stimulation interval as the other cells in this paper?

To clarify the discussion, we divided the original figure into two separate panels (new Fig. S10, 11). First, new Fig. S10 presents a rare case in which the increase and decrease of the cAMP signal occur in the same direction. We explained the underlying reason for this phenomenon using the pipette assay (Fig. 4): specifically, we interpret this as the cell responding to a signal arriving from a direction different from the one defined by its established polarity. In line with this interpretation, new Fig. S11 illustrates how the cell gradually aligns its polarity with the incoming signal from another direction. We revised the corresponding text to make this explanation clearer, as follows:

In some giant cells, the sites of increase and decrease in cAMP levels coincided in some cases (Supplementary Fig. 10 and Supplementary Video 4). Furthermore, in the cell repeatedly stimulated by spontaneous cAMP signals, the site of cAMP decrease gradually shifted toward the rear end of the cell over time (Supplementary Fig. 11 and Supplementary Video 4–7), suggesting that the incoming signal started to arrive from a direction unaligned with the previously established polarity of the cell.

Figure 4b:

It will be helpful to indicate front and rear in Fig. 3b. How does the heatmap and cAMP kinetics look like in front and rear of cell in (c) and (d). This will give a clearer understanding of how cAMP

signalling is spatiotemporally regulated with respect to the site of extracellular stimulation

To improve clarity, we enhanced the labeling of the Front (F) and Rear (R) in Fig. 3 as well as in other figures that display clear front–rear cell polarity. Furthermore, we added heatmap-based directional analyses for both cAMP increases and decreases (Fig. 3h and Fig. S3).

Figure 5:

Is it possible to show a control cell of normal size? In addition, cells may induce larger vesicles (several μm in size) due to cellular stress, independent of their sizes. The authors should comment on this possibility. In addition, there is no statistics of this phenotype.

In electron microscopy, front–rear labeling is challenging, and the anterior–posterior polarity of normal-sized cells is often unclear. For the same reason, although asymmetric vesicle localization can be observed in other cells as well, only a limited number of cells exhibit polarity clear enough for reliable interpretation, making statistical evaluation difficult. As the reviewer pointed out, the presence of large vesicles is particularly prominent in the enlarged cells. We have therefore added the following description to the main text:

While the large vesicles may be characteristic of the giant cells due to their size, the clusters of smaller vesicles appear to fall within a reasonable size range even in normal-sized cells.

Figure 6: Ca^{2+} coupling

The biphasic Ca^{2+} response to cAMP pulses is intriguing but requires clarification of how cAMP and Ca^{2+} signals are coupled inside of the cell.

The reviewer's point is important. We attempted simultaneous imaging of cAMP and Ca^{2+} signals (see attached Fig. Ex1). In the simultaneous measurements, a red fluorescent probe (jRGECO1a) was required for Ca^{2+} , whereas the individual Ca^{2+} measurements shown in the manuscript use a green probe (GCaMP6s). Because the red probe has lower sensitivity than that of GCaMP6s (in terms of both K_d and dynamic range), the biphasic Ca^{2+} responses were rarely detected, and the temporal coordination between the two signals remained unclear. We therefore consider this an issue for future investigation.

Furthermore, the purpose of the calcium imaging in this study is primarily to demonstrate that other signaling events can also be monitored in the giant cells. To clarify this point, we added the following statement to the main text.

To address this and to demonstrate the versatility of signal measurements in giant cells, we expressed the genetically encoded Ca^{2+} indicator GCaMP6s in wild-type AX2 cells and induced giant cell formation to enable high-resolution

analysis of intracellular Ca^{2+} dynamics.

Fig. Ex1: Simultaneous measurement of cAMP and Ca^{2+} signals with Flamindo2 and jRGECO1a.

Figure 7: Ca^{2+} –actin relationship

To be able to capture intracellular Ca^{2+} spatial dynamics and to couple it with F-actin waves is interesting and indeed limited in normal sized cells. There are however better ways to illustrate their point. It is not immediately clear how the dynamics of Ca^{2+} and F-actin are coordinated. The kymograph did not show a clear overlap. Figure 7d is barely visible. It will be helpful to overlay the time series of normalized Ca^{2+} and F-actin signal intensities from Fig I and J (but split front and back in two figures) to better understand how they are coupled with each other.

Following the reviewer's recommendation, we improved the contrast of the kymograph (new Fig. 7d) and incorporated an overlay of the Ca^{2+} signal and actin dynamics time-series data (new Fig. 7l).

Minor Comments

- Scale bar is missing on Figure 3a, 4c, 4e, 5a, 6c, 7g.

We added scales bars to the listed figures.

- In the lookup table for Fig2, Fig 4, red is low, blue is high; in Fig 6a, red is high, blue is low. Please make it consistent. The conventional view is that warm color is high.

We have revised both the figures and the corresponding movies.

- Figure 4 legend Line 797/Line 801: time is shown in the upper left, not right.

We corrected the position of the timestamp in the legend.

- Figure 6b, 7b,d,f: indicate front and rear on kymographs.

We indicated front and rear on the kymographs.

Responses to reviewers

Reviewers' comments are in blue below, with our responses in black.

Reviewer #1:

I have evaluated the corrected manuscript and I am happy that the author's have addressed my concerns in the majority of cases. The presentation and speed of the movies in particular is a big improvement. I only have 2 outstanding issues, one major, one minor:

Major point (relating to my original comment regarding "acceleration phenomenon"):

Although we examined more cell datasets than those shown in the manuscript, this particular case is relatively uncommon, though similar patterns can occasionally be detected. Therefore, we softened the wording as follows:

“Notably, such acceleration was not consistently observed across all recorded cells, implying that propagation may vary to some extent (Supplementary Fig. 4, 5).”

I still fundamentally disagree with this and find the authors' response inadequate. From the data presented, 1 cell out of 6 cells exhibit “acceleration”. This implies the majority of the time there isn't acceleration and that any pattern could be (or even should be considered) random. So, I do not believe anything can be concluded or proposed based on these data and any claim or discussion about acceleration should be removed. To put it bluntly, you can't just pick one cell you saw something interesting with and try and make a claim out of it and say that all the other cells represent “some” variability. Stating that you haven't presented all analysed data isn't reassuring either. If the authors want to claim there is something to this (even softened as is), I'd argue they need to present data proving it is more common than not for it to be considered a "pattern" or, at the very least a minimum of $n=3$ observations to be considered a consistent feature.

We sincerely thank the reviewer for the careful re-evaluation of our revised manuscript and for the constructive feedback. We appreciate the reviewer's strong concerns regarding the interpretation of the “acceleration phenomenon,” and we fully understand the need for caution when drawing conclusions from limited observations.

In response, we have deleted all mentions of acceleration from the Results section of the manuscript.

Minor point:

“Consistently with this, we observed both responses with clear front–rear polarity and responses without such polarity (Fig. 6, 7).”

I don't find this newly added sentence very clear.

To improve clarity, we propose revising this sentence to:

“Consistently with this, some responses displayed clear front–rear differences, whereas others showed more spatially uniform patterns (Fig. 6, 7).”

I feel very strongly about the first point. Unless I have misunderstood the data presented or the authors' can provide further examples, I do not believe this can be included in the paper as is.

As suggested by the reviewer, we have removed all descriptions related to the first point from the manuscript.